# Regret Minimization in Stackelberg Games with Side Information

**Keegan Harris**
School of Computer Science
Carnegie Mellon University
Pittsburgh, PA 15213
keeganh@cs.cmu.edu

**Zhiwei Steven Wu**
School of Computer Science
Carnegie Mellon University
Pittsburgh, PA 15213
zhiweiw@cs.cmu.edu

**Maria-Florina Balcan**
School of Computer Science
Carnegie Mellon University
Pittsburgh, PA 15213
ninamf@cs.cmu.edu

## Abstract

Algorithms for playing in Stackelberg games have been deployed in real-world domains including airport security, anti-poaching efforts, and cyber-crime prevention. However, these algorithms often fail to take into consideration the additional information available to each player (e.g. traffic patterns, weather conditions, network congestion), which may significantly affect both players' optimal strategies. We formalize such settings as *Stackelberg games with side information*, in which both players observe an external *context* before playing. The leader commits to a (context-dependent) strategy, and the follower best-responds to both the leader's strategy and the context. We focus on the online setting in which a sequence of followers arrive over time, and the context may change from round-to-round. In sharp contrast to the non-contextual version, we show that it is impossible for the leader to achieve no-regret in the full adversarial setting. Motivated by this result, we show that no-regret learning is possible in two natural relaxations: the setting in which the sequence of followers is chosen stochastically and the sequence of contexts is adversarial, and the setting in which contexts are stochastic and follower types are adversarial.

## 1 Introduction

A *Stackelberg game* [30, 31] is a strategic interaction between two utility-maximizing players in which one player (the *leader*) is able to *commit* to a (possibly mixed) strategy before the other player (the *follower*) takes an action. While Stackelberg's original formulation was used to model economic competition between firms, Stackelberg games have been used to study a wide range of topics in computing ranging from incentives in algorithmic decision-making [15] to radio spectrum utilization [32]. Perhaps the most successful application of Stackelberg games to solve real-world problems is in the field of security, where the analysis of *Stackelberg security games* has led to new methods in domains such as passenger screening at airports [8], wildlife protection efforts in conservation areas [10], the deployment of Federal Air Marshals on board commercial flights [18], and patrol boat schedules for the United States Coast Guard [1].[1]

However in many real-world settings which are typically modeled as Stackelberg games, the payoffs of the players often depend on additional *contextual information* which is not captured by the Stackelberg game framework. For example, in airport security the severity of an attack (as well as the "benefit" of a successful attack to the attacker) depends on factors such as the arrival and departure city of a flight, the whether there are VIP passengers on board, and the amount of valuable cargo on the aircraft. Additionally, there may be information in the time leading up to the attack attempt which may help the security service determine the type of attack which is coming [17]. For instance, in wildlife protection settings factors such as the weather and time of year may make certain species of

---

[1]See [27, 19, 2] for an overview of other application domains for Stackelberg security games.

wildlife easier or harder to defend from poaching, and information such as the location of tire tracks may provide context about which animals are being targeted. As a result, the optimal strategy of both the leader and the follower may change significantly depending on the side information available.

**Overview of our results.** In order to capture the additional information that the leader and follower may have at their disposal, we formalize such settings as *Stackelberg games with side information*. Specifically, we consider a setting in which a leader interacts with a sequence of followers in an online fashion. At each time-step, the leader gets to see payoff-relevant information about the current round in the form of a *context*. After observing the context, the leader commits to a mixed strategy over a finite set of actions, and the follower best-responds to both (1) the leader's strategy and (2) the context in order to maximize their utility. We allow the follower in each round to be one of $K$ *types*. Each follower type corresponds to a different mapping from contexts, leader strategies, and follower actions to utilities. While the leader may observe the context before committing to their mixed strategy, they do not get to observe the follower's type until after the round is over. Under this setting, the goal of the leader is to minimize their *regret* with respect to the best *policy* (i.e. the best mapping from contexts to mixed strategies) in hindsight, with respect to the realized sequence of followers and contexts.

We show that in the fully adversarial setting (i.e. the setting in which both the sequence of contexts and follower types is chosen by an adversary), no-regret learning is not possible, even when the policy class is highly structured. We show this via a reduction from the problem of online linear thresholding, for which it is known that no no-regret learning algorithm exists. Motivated by this impossibility result, we study two natural relaxations: (1) a setting in which the sequence of contexts is chosen by an adversary and the sequence of follower types is chosen stochastically, and (2) a setting in which the sequence of contexts is chosen stochastically and the sequence of follower types is chosen by an adversary.

In the stochastic follower setting we show that the greedy algorithm (Algorithm 1), which estimates the leader's expected utility for the given context and plays the mixed strategy which maximizes their estimate, achieves no-regret as long as the total variation distance between their estimate and the true distribution is decreasing with time. We then show how to instantiate the leader's estimation procedure so that the regret of Algorithm 1 is $O(\min\{K, A_f\}\sqrt{T\log(T)})$, where $T$ is the time horizon, $K$ is the number of follower types, and $A_f$ is the number of follower actions. In the stochastic context setting, we show the leader can obtain $O(\sqrt{KT\log(T)} + K)$ regret by playing Hedge over a finite set of policies (Algorithm 2). An important intermediate result in both settings is that it is (nearly) without loss of generality to consider leader policies which map to a finite set of mixed strategies $\mathcal{E}_{\mathbf{z}}$, given context $\mathbf{z}$.[2]

Next, we extend our algorithms for both types of adversary to the setting in which the leader does not get to observe the follower's type after each round, but instead only gets to observe their action. We refer to this type of feedback as *bandit feedback*. Both of our extensions to bandit feedback make use of the notion of a *barycentric spanner* [4], a special basis under which bounded loss estimators may be obtained for all leader mixed strategies. In the bandit stochastic follower setting, we use the fact that in addition to being bounded, a loss estimator constructed using a barycentric spanner has low variance, in order to show that a natural extension of our greedy algorithm obtains $\tilde{O}(T^{2/3})$ regret. We also make use of barycentric spanners in the (bandit) stochastic context setting, albeit in a different way. Here, our algorithm proceeds by splitting the time horizon into blocks, and using a barycentric spanner to estimate the leader's utility from playing according to a set of special policies in each block. We then play Hedge over these policies to obtain $\tilde{O}(T^{2/3})$ leader regret.[3] See Table 1 for a summary of our results.

**Related work.** Letchford et al. [21] consider the problem of learning the leader's optimal mixed strategy in the repeated Stackelberg game setting against a perfectly rational follower with an unknown payoff matrix. Peng et al. [23] study the same setting as Letchford et al. [21]. They provide improved rates and prove nearly-matching lower bounds. Learning algorithms to recover the leader's optimal mixed strategy have also been studied in Stackelberg security games [5, 7, 26, 6].

Our work builds off of several results established for online learning in (non-contextual) Stackelberg games in Balcan et al. [5]. In particular, our results in Section 4.2 and Appendix C.2 may be viewed as a generalization of their results to the setting in which the payoffs of both players depend on an external context. Roughly speaking, Balcan et al. [5] show that it without loss to play Hedge over a finite set

---

[2]Specifically, a leader who is restricted to playing such policies incurs negligible additional regret.

[3]This is similar to how Balcan et al. [5] use barycentric spanners to obtain no-regret in the non-contextual setting, although more care must be taken to handle the side information present in our setting.

| | Full Feedback | Bandit Feedback |
| --- | --- | --- |
| Fully Adversarial | $\Omega(T)$ (Section 3) | $\Omega(T)$ (Section 3) |
| Stochastic Followers, Adversarial Contexts | $O\left(\min\{K, A_f\}\sqrt{T\log T}\right)$ (Section 4.1) | $O\left(K^{2/3}A_f^{2/3}T^{2/3}\log^{1/3}T\right)$ (Appendix C.1) |
| Stochastic Contexts, Adversarial Followers | $O\left(\sqrt{KT\log T}+K\right)$ (Section 4.2) | $O\left(KA_f^{1/3}T^{2/3}\log^{1/3}T\right)$ (Appendix C.2) |

Table 1: Summary of our results. Under bandit feedback, we consider a relaxed setting in which only the leader's utility depends on the side information.

of *mixed strategies* in order to obtain no-regret against an adversarially-chosen sequence of follower types. In order to handle the additional side information available in our setting, we instead play Hedge over a finite set of *policies*, each of which map to a finite set of (context-dependent) mixed strategies. However, the discretization argument is more nuanced in our setting. In particular, it is not without loss of generality to consider a finite set of policies. As a result, we need to bound the additional regret incurred by the leader due to the discretization. More recent work on learning in Stackelberg games provides improved regret rates in the full feedback [9] and bandit feedback [6] settings, and considers the effects of non-myopic followers [13] and followers who respond to calibrated predictions [14].

Lauffer et al. [20] study a Stackelberg game setting in which there is an underlying (probabilistic) state space which affects the leader's rewards, and there is a single (unknown) follower type. In contrast, we study a setting in which the sequence of follower types and/or contexts may be chosen adversarially. Sessa et al. [25] study a repeated game setting in which the players receive additional information (i.e. a context) at each round, much like in our setting. However, their focus is on repeated normal-form games, which require different tools and techniques to analyze compared to the repeated Stackelberg game setting we consider. Other work has also considered repeated normal-form games which change over time in different ways. In particular, Zhang et al. [33], Anagnostides et al. [3] study learning dynamics in time-varying game settings, and Harris et al. [16] study a meta-learning setting in which the game being played changes after a fixed number of rounds.

Finally, our problem may be viewed is a special case of the contextual bandit setting with adversarially-chosen utilities [28, 29, 24], where the learner gets to observe "extra information" in the form of the follower's type (Section 4) or the follower's action (Section 5). However, there is much to gain from taking advantage of the additional information and structure that is present in our setting. Besides having generally worse regret rates, another reason not to use off-the-shelf adversarial contextual bandit algorithms in our setting is that they typically require either (1) the learner to know the set of contexts they will face beforehand (the *transductive* setting; Syrgkanis et al. [28, 29], Rakhlin and Sridharan [24]) or (2) for there to exist a small set of contexts such that any two policies behave differently on at least one context in the set (the *small separator* setting; Syrgkanis et al. [28]). We require no such assumptions.

## 2 Setting and background

**Notation.** We use $[N] := \{1, \ldots, N\}$ to denote the set of natural numbers up to and including $N \in \mathbb{N}$ and $\mathrm{cl}(\mathcal{P})$ to denote the closure of the set $\mathcal{P}$. $\mathbf{x}[a]$ denotes the $a$-th component of vector $\mathbf{x}$, and $\Delta(\mathcal{A})$ denotes the probability simplex over the set $\mathcal{A}$. $\mathrm{TV}(\mathbf{p}, \mathbf{q}) = \frac{1}{2}\int |p(x) - q(x)|dx$ is the total variation distance between distributions $\mathbf{p}$ and $\mathbf{q}$, and $\mathbb{E}_t[x] = \mathbb{E}[x|\mathcal{F}_t]$ is shorthand for the expected value of the random variable $x$, conditioned on the filtration up to (but not including) time $t$. All proofs may be found in the Appendix. Finally, while we present our results for general Stackelberg games with side information, our results are readily applicable to the special case of Stackelberg *security* games with side information.

**Our setting.** We consider a game between a leader and a sequence of followers. In each round $t \in [T]$, Nature reveals a context $\mathbf{z}_t \in \mathcal{Z} \subseteq \mathbb{R}^d$ to both players.[4] The leader moves first by playing some mixed strategy $\mathbf{x}_t \in \mathcal{X} \subseteq \mathbb{R}^A$ over a set of (finite) leader actions $\mathcal{A}$, i.e., $\mathbf{x}_t \in \Delta(\mathcal{A})$. The size of $\mathcal{A}$

---

[4]E.g. in airport security, $\mathbf{z}_t$ may contain information about arrival and departure times, number of passengers, valuable cargo, etc. In cyber-defense, $\mathbf{z}_t$ may be a list of network traffic statistics.

is $A := |\mathcal{A}|$. Having observed the leader's mixed strategy, the follower *best-responds* to both $\mathbf{x}_t$ and $\mathbf{z}_t$ by playing some action $a_f \in \mathcal{A}_f$, where $\mathcal{A}_f$ is the (finite) set of follower actions and $A_f := |\mathcal{A}_f|$.

**Definition 2.1** (Follower Best-Response)**.** *Follower $f$'s best-response to context $\mathbf{z}$ and mixed strategy $\mathbf{x}$ is $b_f(\mathbf{z}, \mathbf{x}) \in \arg\max_{a_f \in \mathcal{A}_f} \sum_{a_l \in \mathcal{A}} \mathbf{x}[a_l] \cdot u_f(\mathbf{z}, a_l, a_f)$, where $u_f : \mathcal{Z} \times \mathcal{A} \times \mathcal{A}_f \to [0, 1]$ is follower $f$'s utility function. In the case of ties, we assume that there is a fixed and known ordering over actions which determines how the follower best-responds, i.e. if $a > a'$ for $a, a' \in \mathcal{A}_f$ then the follower will break ties between $a$ and $a'$ in favor of $a$.*[5]

We allow for the follower in round $t$ (denoted by $f_t$) to be one of $K \geq 1$ *follower types* $\{\alpha^{(1)}, \ldots, \alpha^{(K)}\}$ (where $K \leq T$). Follower type $\alpha^{(i)}$ is characterized by utility function $u_{\alpha^{(i)}} : \mathcal{Z} \times \mathcal{A} \times \mathcal{A}_f \to [0, 1]$, i.e. given a context $\mathbf{z}$, leader action $a_l$, and follower action $a_f$, a follower of type $\alpha^{(i)}$ would receive utility $u_{\alpha^{(i)}}(\mathbf{z}, a_l, a_f)$. We assume that the set of all possible follower types and their utility functions are known to the leader, but that the follower's type at round $t$ is not revealed to the leader until *after* the round is over. We denote the leader's utility function by $u : \mathcal{Z} \times \mathcal{A} \times \mathcal{A}_f \to [0, 1]$ and assume it is known to the leader. We often use the shorthand $u(\mathbf{z}, \mathbf{x}, a_f) = \sum_{a_l \in \mathcal{A}} \mathbf{x}[a_l] \cdot u(\mathbf{z}, a_l, a_f)$ to denote the leader's expected utility of playing mixed strategy $\mathbf{x}$ under context $\mathbf{z}$ against follower action $a_f$. Follower $f_t$'s expected utility $u_{f_t}(\mathbf{z}, \mathbf{x}, a_f)$ is defined analogously.

A leader *policy* $\pi : \mathcal{Z} \to \mathcal{X}$ is a (possibly random) mapping from contexts to mixed strategies. If the leader using policy $\pi_t$ in round $t$ and observes context $\mathbf{z}_t$, their strategy $\mathbf{x}_t$ is given by $\mathbf{x}_t \sim \pi_t(\mathbf{z}_t)$.

**Definition 2.2** (Optimal Policy)**.** *Given a sequence of followers $f_1, \ldots, f_T$ and contexts $\mathbf{z}_1, \ldots, \mathbf{z}_T$, the strategy given by the leader's optimal-in-hindsight policy for context $\mathbf{z}$ is $\pi^*(\mathbf{z}) \in \arg\max_{\mathbf{x} \in \mathcal{X}} \sum_{t=1}^{T} u(\mathbf{z}, \mathbf{x}, b_{f_t}(\mathbf{z}, \mathbf{x})) \cdot \mathbb{1}\{\mathbf{z}_t = \mathbf{z}\}$.*

We measure the leader's performance against the optimal policy via the notion of *contextual Stackelberg regret* (*regret* for short).

**Definition 2.3** (Contextual Stackelberg Regret)**.** *Given a sequence of followers $f_1, \ldots, f_T$ and contexts $\mathbf{z}_1, \ldots, \mathbf{z}_T$, the leader's contextual Stackelberg regret is $R(T) := \sum_{t=1}^{T} u(\mathbf{z}_t, \pi^*(\mathbf{z}_t), b_{f_t}(\mathbf{z}_t, \pi^*(\mathbf{z}_t))) - u(\mathbf{z}_t, \mathbf{x}_t, b_{f_t}(\mathbf{z}_t, \mathbf{x}_t))$, where $\mathbf{x}_1, \ldots, \mathbf{x}_T$ is the sequence of mixed strategies played by the leader.*

If an algorithm achieves regret $R(T) = o(T)$, we say that it is a *no-regret* algorithm. We consider three ways in which Nature can select the sequence of contexts/followers:

1. If the sequence of contexts (resp. follower types) are drawn i.i.d. from some fixed distribution, we say that the sequence of contexts (resp. follower types) are chosen *stochastically*.

2. If Nature chooses the sequence of contexts (resp. follower types) before the first round in order to harm the leader (possibly using knowledge of the leader's algorithm), we say that the sequence of contexts (resp. follower types) are chosen by a *non-adaptive adversary*.

3. If Nature chooses context $\mathbf{z}_t$ (resp. follower $f_t$) before round $t$ in order to harm the leader (possibly using knowledge of the leader's algorithm and the outcomes of the prior $t-1$ rounds), we say that the sequence of contexts (resp. follower types) are chosen by an *adaptive adversary*.

Our impossibility results in Section 3 hold when both the sequence of contexts *and* the sequence of follower types are chosen by either type of adversary. Our positive results in Section 4 hold when either the sequence of contexts *or* the sequence of follower types are chosen by either type of adversary (and the other sequence is chosen stochastically). Our extension to bandit feedback (Section 5, where the leader only gets to observe the follower's best-response instead of their type) holds whenever one sequence is chosen by a non-adaptive adversary and the other sequence is chosen stochastically.

## 3 On the impossibility of fully adversarial no-regret learning

We begin with a negative result: no-regret learning is not possible in the setting of Section 2 if the sequence of contexts and the sequence of followers is chosen by an adversary. While this is not necessarily surprising given that Definition 2.3 allows for the optimal policy $\pi^*$ to be arbitrarily

---

[5]It is without loss of generality to assume that the follower's best-response is a pure strategy.

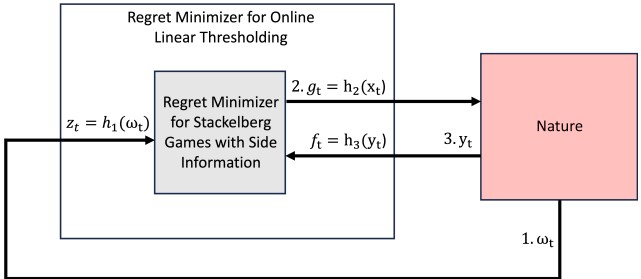

Figure 1: Summary of our reduction from the online linear thresholding problem. At time $t \in [T]$, (1.) the learner observes a point $\omega_t$, (2.) the learner takes a guess $g_t$, and (3.) the learner observes the true label $y_t$. Given a regret minimizer for our setting, we show how to use it in a black-box way (by constructing functions $h_1$, $h_2$, $h_3$) to achieve no-regret in the online linear thresholding problem.

complex, we show that this result holds *even when the policy class to which $\pi^*$ belongs is highly structured*. We show this via a reduction to the online linear thresholding problem, for which it is known that no-regret learning is impossible.

**Online linear thresholding.** The online linear thresholding problem is a repeated two-player game between a learner and an adversary. Before the first round, an adversary chooses a *cutoff* $s \in [0, 1]$ which is *unknown* to the learner. In each round, the adversary chooses a point $\omega_t \in [0, 1]$ and reveals it to the learner. $\omega_t$ is assigned label $y_t = 1$ if $\omega_t > s$ and label $y_t = -1$ otherwise. Given $\omega_t$, the learner makes a *guess* $g_t \in [0, 1]$ (the probability they place on $y_t = 1$), and receives utility $u_{\mathrm{OLT}}(\omega_t, g_t) = g_t \cdot \mathbb{1}\{y_t = 1\} + (1 - g_t) \cdot \mathbb{1}\{y_t = -1\}$. The learner gets to observe $y_t$ after round $t$ is over. The learner's policy $\pi_t : [0, 1] \to [0, 1]$ is a mapping from points in $[0, 1]$ to guesses in $[0, 1]$. The optimal policy $\pi^*$ makes guess $\pi^*(\omega_t) = 1$ if $\omega_t > s$ and $\pi^*(\omega_t) = 0$ otherwise. The learner's regret after $T$ rounds is given by $R_{\mathrm{OLT}}(T) = T - \sum_{t=1}^T u_{\mathrm{OLT}}(\omega_t, g_t)$, since the optimal policy achieves utility 1 in every round. In order to prove a lower bound on contextual Stackelberg regret in our setting, we make use of the following well-known lower bound on regret in the online linear thresholding setting (see e.g. [12]).

**Lemma 3.1.** *Any algorithm suffers regret $R_{\mathrm{OLT}}(T) = \Omega(T)$ in the online linear thresholding problem when the sequence of points $\omega_1, \dots, \omega_T$ is chosen by an adversary.*

**Theorem 3.2.** *If an adversary can choose both the sequence of contexts $\mathbf{z}_1, \dots, \mathbf{z}_T$ and the sequence of followers $f_1, \dots, f_T$, no algorithm can achieve better than $\Omega(T)$ contextual Stackelberg regret in expectation over the internal randomness of the algorithm, even when $\pi^*$ is restricted to come from the set of linear thresholding functions.*

The reduction from online linear thresholding proceeds by creating an instance of our setting such that the sequence of contexts $z_1, \dots, z_T$ correspond to the sequence of points $\omega_1, \dots, \omega_T$ encountered by the learner, and the sequence of follower types $f_1, \dots, f_T$ correspond to the sequence of labels $y_1, \dots, y_T$. We then show that a no-regret algorithm in the online thresholding problem can be obtained by using an algorithm which minimizes contextual Stackelberg regret on the constructed game instance as a black box. However this is a contradiction, since by Lemma 3.1 the online thresholding problem is not online learnable by any algorithm. See Figure 1 for a visualization of our reduction.

Intuitively, this reduction works because the adversary can "hide" information about the follower's type $f_t$ in the context $z_t$. However, there exists a family of problem instances in which learning this relationship between contexts and follower types as hard as learning the threshold in the online linear thresholding problem, for which no no-regret learning algorithm exists by Lemma 3.1.

## 4 Limiting the power of the adversary

Motivated by the impossibility result of Section 3, we study two natural relaxations of the fully adversarial setting: one in which the sequence of followers is chosen stochastically but the contexts are chosen adversarially (Section 4.1) and one in which the sequence of contexts is chosen stochastically but followers are chosen adversarially (Section 4.2). In both settings we allow the adversary to be adaptive.

An important structural results for both Section 4.1 and Section 4.2 is that for any context $\mathbf{z} \in \mathcal{Z}$, the leader incurs only negligible regret by restricting themselves to policies which map to mixed

strategies in some finite (and computable) set $\mathcal{E}_{\mathbf{z}}$. In order to state this result formally, we need to introduce the notion of a *contextual best-response region*, which is a generalization of the notion of a best-response region in (non-contextual) Stackelberg games (e.g. [21, 5]).

**Definition 4.1** (Contextual Follower Best-Response Region). *For follower type $\alpha^{(i)}$, follower action $a_f \in \mathcal{A}_f$, and context $\mathbf{z} \in \mathcal{Z}$, let $\mathcal{X}_{\mathbf{z}}(\alpha^{(i)}, a_f) \subseteq \mathcal{X}$ denote the set of all leader mixed strategies such that a follower of type $\alpha^{(i)}$ best-responds to all $\mathbf{x} \in \mathcal{X}_{\mathbf{z}}(\alpha^{(i)}, a_f)$ by playing action $a_f$ under context $\mathbf{z}$, i.e., $\mathcal{X}_{\mathbf{z}}(\alpha^{(i)}, a_f) = \{\mathbf{x} \in \mathcal{X} : b_{\alpha^{(i)}}(\mathbf{z}, \mathbf{x}) = a_f\}$.*

**Definition 4.2** (Contextual Best-Response Region). *For a given function $\sigma : \{\alpha^{(1)}, \ldots, \alpha^{(K)}\} \to \mathcal{A}_f$, let $\mathcal{X}_{\mathbf{z}}(\sigma)$ denote the set of all leader mixed strategies such that under context $\mathbf{z}$, a follower of type $\alpha^{(i)}$ plays action $\sigma(\alpha^{(i)})$ for all $i \in [K]$, i.e. $\mathcal{X}_{\mathbf{z}}(\sigma) = \cap_{i \in [K]} \mathcal{X}_{\mathbf{z}}(\alpha^{(i)}, \sigma(\alpha^{(i)}))$.*

For a fixed contextual best-response region $\mathcal{X}_{\mathbf{z}}(\sigma)$, we refer to the corresponding $\sigma$ as the *best-response function* for region $\mathcal{X}_{\mathbf{z}}(\sigma)$, as it maps each follower type to its best-response for every leader strategy $\mathbf{x} \in \mathcal{X}_{\mathbf{z}}(\sigma)$. We sometimes use $\sigma^{(\mathbf{z}, \mathbf{x})}$ to refer to the best-response function associated with mixed strategy $\mathbf{x}$ under context $\mathbf{z}$, and we use $\Sigma_{\mathbf{z}}$ to refer to the set of all best-response functions under context $\mathbf{z}$. Note that $|\Sigma_{\mathbf{z}}| \leq A_f^K$ for any context $\mathbf{z} \in \mathcal{Z}$. This gives us an upper-bound on the number of best-response regions for a given context.

One useful property of all contextual best-response regions is that they are convex and bounded polytopes. To see this, observe that every contextual follower best-response region (and therefore every contextual best-response region) is (1) a subset of $\Delta^d$ and (2) the intersection of finitely-many half-spaces. While every $\mathcal{X}_{\mathbf{z}}(\sigma)$ is convex and bounded, it is not necessarily closed. If every contextual best-response region were closed, it would be without loss of generality for the leader to restrict themselves to the set of policies which map every context to an extreme point of some contextual best-response region. In what follows, we show that the leader does not "lose too much" (as measured by regret) by restricting themselves to policies which map to some *approximate* extreme point of a contextual best-response region.

**Definition 4.3** ($\delta$-approximate extreme points). *Fix a context $\mathbf{z} \in \mathcal{Z}$ and consider the set of all non-empty contextual best-response regions. For $\delta > 0$, $\mathcal{E}_{\mathbf{z}}(\delta)$ is the set of leader mixed strategies such that for all best-response functions $\sigma$ and any $\mathbf{x} \in \Delta(\mathcal{A}_l)$ that is an extreme point of $cl(\mathcal{X}_{\mathbf{z}}(\sigma))$, $\mathbf{x} \in \mathcal{E}_{\mathbf{z}}(\delta)$ if $\mathbf{x} \in \mathcal{X}_{\mathbf{z}}(\sigma)$. Otherwise there is some $\mathbf{x}' \in \mathcal{E}_{\mathbf{z}}(\delta)$ such that $\mathbf{x}' \in \mathcal{X}_{\mathbf{z}}(\sigma)$ and $\|\mathbf{x}' - \mathbf{x}\|_1 \leq \delta$.*

Note that Definition 4.3 is constructive. We set $\delta = \mathcal{O}(\frac{1}{T})$ so that the additional regret from only considering policies which map to points in $\cup_{\mathbf{z} \in \mathcal{Z}} \mathcal{E}_{\mathbf{z}}(\delta)$ is negligible. As a result, we use the shorthand $\mathcal{E}_{\mathbf{z}} := \mathcal{E}_{\mathbf{z}}(\delta)$ throughout the sequel. The following lemma is a generalization of Lemma 4.3 in Balcan et al. [5] to our setting, and its proof uses similar techniques from convex analysis.

**Lemma 4.4.** *For any sequence of followers $f_1, \ldots f_T$ and any leader policy $\pi$, there exists a policy $\pi^{(\mathcal{E})} : \mathcal{Z} \to \cup_{\mathbf{z} \in \mathcal{Z}} \mathcal{E}_{\mathbf{z}}$ that, when given context $\mathbf{z}$, plays a mixed strategy in $\mathcal{E}_{\mathbf{z}}$ and guarantees that $\sum_{t=1}^{T} u(\mathbf{z}_t, \pi(\mathbf{z}_t), b_{f_t}(\mathbf{z}_t, \pi(\mathbf{z}_t))) - u(\mathbf{z}_t, \pi^{(\mathcal{E})}(\mathbf{z}_t), b_{f_t}(\mathbf{z}_t, \pi^{(\mathcal{E})}(\mathbf{z}_t))) \leq 1$. Moreover, the same result holds in expectation over any distribution over follower types $\mathcal{F}$.*

Since we do not restrict the context space to be finite, the leader cannot pre-compute $\mathcal{E}_{\mathbf{z}}$ for every $\mathbf{z} \in \mathcal{Z}$ before the game begins. Instead, they can compute $\mathcal{E}_{\mathbf{z}_t}$ in round $t$ before they commit to their mixed strategy. While $\mathcal{E}_{\mathbf{z}_t}$ is computatable, it may be exponentially large in $A_f$ and $K$. However this is to be expected as Li et al. [22] show that in its general form, solving the non-contextual version of the online Stackelberg game problem is NP-Hard.

## 4.1 Stochastic follower types and adversarial contexts

In this setting we allow the sequence of contexts to be chosen by an adversary, but we restrict the sequence of followers to be sampled i.i.d. from some (unknown) distribution over follower types $\mathcal{F}$. When picking context $\mathbf{z}_t$, we allow the adversary to have knowledge of $\mathcal{F}$ and $f_1, \ldots, f_{t-1}$, but not $f_t$. Under this relaxation, our measure of algorithm performance is *expected* contextual Stackelberg regret, where the expectation is taken over the randomness in the distribution over follower types.

**Definition 4.5** (Expected Contextual Stackelberg Regret). *Given a distribution over followers $\mathcal{F}$ and a sequence of contexts $\mathbf{z}_1, \ldots, \mathbf{z}_T$, the leader's expected contextual Stackeleberg regret is $\mathbb{E}[R(T)] := \mathbb{E}_{f_1, \ldots, f_T \sim \mathcal{F}}[\sum_{t=1}^{T} u(\mathbf{z}_t, \pi^*(\mathbf{z}_t), b_{f_t}(\mathbf{z}_t, \pi^*(\mathbf{z}_t))) - u(\mathbf{z}_t, \mathbf{x}_t, b_{f_t}(\mathbf{z}_t, \mathbf{x}_t))]$, where $\pi^*$ is the optimal policy given knowledge of $\mathbf{z}_1, \ldots, \mathbf{z}_T$ and $\mathcal{F}$.*

**Algorithm 1:** Learning with stochastic follower types: full feedback

---

**Input:** $\widehat{\mathbf{p}}_1$

**for** $t = 1, \ldots, T$ **do**

    Observe $\mathbf{z}_t$, commit to $\mathbf{x}_t = \pi_t(\mathbf{z}_t) = \arg\max_{\mathbf{x} \in \mathcal{E}_{\mathbf{z}_t}} \widehat{\mathbb{E}}_t[u(\mathbf{z}_t, \mathbf{x}, b_{f_t}(\mathbf{z}_t, \mathbf{x})]$ (Equation (1)).

    Receive utility $u(\mathbf{z}_t, a_l, b_{f_t}(\mathbf{z}_t, \mathbf{x}_t))$ where $a_l \sim \mathbf{x}_t$, and observe follower type $f_t$.

    Update $\widehat{\mathbf{p}}_t \to \widehat{\mathbf{p}}_{t+1}$

**end**

---

Under this setting, the utility for policy $\pi$ may be written as
$$\mathbb{E}_{f_1,\ldots,f_T \sim \mathcal{F}}\left[\sum_{t=1}^T u(\mathbf{z}_t, \pi(\mathbf{z}_t), b_{f_t}(\mathbf{z}_t, \pi(\mathbf{z}_t)))\right] = \sum_{t=1}^T \mathbb{E}_{f_1,\ldots,f_{t-1} \sim \mathcal{F}}[\mathbb{E}_t[u(\mathbf{z}_t, \pi(\mathbf{z}_t), b_{f_t}(\mathbf{z}_t, \pi(\mathbf{z}_t)))]].$$
Our algorithm (Algorithm 1) proceeds by estimating the inner expectation $\mathbb{E}_t[u(\mathbf{z}_t, \pi(\mathbf{z}_t), b_{f_t}(\mathbf{z}_t, \pi(\mathbf{z}_t)))]$ as

$$\widehat{\mathbb{E}}_t[u(\mathbf{z}_t, \pi(\mathbf{z}_t), b_{f_t}(\mathbf{z}_t, \pi(\mathbf{z}_t)))] := \int u(\mathbf{z}_t, \pi(\mathbf{z}_t), a_f) d\widehat{p}_t(b_{f_t}(\mathbf{z}_t, \pi(\mathbf{z}_t)) = a_f) \qquad (1)$$

and acting greedily with respect to our estimate. Here $\widehat{p}_t(b_{f_t}(\mathbf{z}_t, \pi(\mathbf{z}_t)) = a_f)$ is the (estimated) probability that the follower's best-response is $a_f$, given context $\mathbf{z}_t$ and leader mixed strategy $\pi(\mathbf{z}_t)$. As we will see, different instantiations of $\widehat{p}_t$ will lead to different regret rates for Algorithm 1. However, before instantiating Algorithm 1 with a specific estimation procedure, we provide a general result which bounds the regret of Algorithm 1 in terms of the total variation distance between the sequence $\{\widehat{p}_t\}_{t \in [T]}$ and the true distribution $p$.

**Theorem 4.6.** *Let* $\mathbf{p}(\mathbf{z}, \mathbf{x}) := [p(b_{f_t}(\mathbf{z}, \mathbf{x}) = a_f)]_{a_f \in \mathcal{A}_f}$ *and* $\widehat{\mathbf{p}}_t(\mathbf{z}, \mathbf{x}) := [\widehat{p}_t(b_{f_t}(\mathbf{z}, \mathbf{x}) = a_f)]_{a_f \in \mathcal{A}_f}$. *The expected contextual Stackelberg regret (Definition 4.5) of Algorithm 1 satisfies*

$$\mathbb{E}[R(T)] \leq 1 + 2\sum_{t=1}^T \mathbb{E}_{f_1,\ldots,f_{t-1}}[\mathrm{TV}(\mathbf{p}(\mathbf{z}_t, \pi^{(\mathcal{E})}(\mathbf{z}_t)), \widehat{\mathbf{p}}_t(\mathbf{z}_t, \pi^{(\mathcal{E})}(\mathbf{z}_t))) + \mathrm{TV}(\mathbf{p}(\mathbf{z}_t, \pi_t(\mathbf{z}_t)), \widehat{\mathbf{p}}_t(\mathbf{z}_t, \pi_t(\mathbf{z}_t)))].$$

Theorem 4.6 shows that the regret of Algorithm 1 is proportional to how well it estimates $\mathbf{p}(\mathbf{z}, \mathbf{x})$ over time (as measured by total variation distance), on (1) the sequence of contexts chosen by the adversary and (2) the sequence of mixed strategies played by Algorithm 1 and the (near-)optimal policy $\pi^{(\mathcal{E})}$. While we instantiate Algorithm 1 in the setting where there are finitely-many follower types and follower actions, Theorem 4.6 opens the door to provide meaningful regret guarantees in settings in which there are infinitely-many follower types and/or follower actions.[6] We now instantiate the estimation procedure in Algorithm 1 in two different ways to get end-to-end regret guarantees. First, the leader can get regret $O(K\sqrt{T \log T})$ by estimating the distribution of follower types directly.

**Corollary 4.7.** *If* $\widehat{\mathbf{p}}_t = \{\widehat{p}_t(f_t = \alpha^{(i)})\}_{i \in [K]}$, $\widehat{p}_{t+1}(f = \alpha^{(i)}) = \frac{1}{t}\sum_{\tau=1}^t \mathbb{1}\{f_\tau = \alpha^{(i)}\}$, *and* $\widehat{p}_1(f = \alpha^{(i)}) = \frac{1}{K}$ *for* $i \in [K]$, *then the regret of Algorithm 1 satisfies* $\mathbb{E}[R(T)] = O(K\sqrt{T \log(T)})$.

The leader can obtain a complementary regret bound of $O(A_f\sqrt{T \log T})$ if they instead estimate the probability that the follower best-responds with action $a_f \in \mathcal{A}_f$, given a particular context $\mathbf{z}$ and leader mixed strategy $\mathbf{x}$.[7] In what follows, we use $\mathbb{1}_{(\sigma(\mathbf{z}, \mathbf{x}) = a_f)} \in \{0, 1\}^K$ to refer to the indicator vector whose $i$-th component is $\mathbb{1}\{\sigma^{(\mathbf{z}, \mathbf{x})}(\alpha^{(i)}) = a_f\}$, i.e. the indicator that a follower of type $\alpha^{(i)}$ best-responds to context $\mathbf{z}$ and mixed strategy $\mathbf{x}$ by playing action $a_f$.

**Corollary 4.8.** *If* $\widehat{\mathbf{p}}_t(\mathbf{z}, \mathbf{x}) = \{\widehat{p}_t(\mathbb{1}_{(\sigma(\mathbf{z}, \mathbf{x}) = a_f)})\}_{a_f \in \mathcal{A}_f}$, $\widehat{p}_{t+1}(\mathbb{1}_{(\sigma(\mathbf{z}, \mathbf{x}) = a)}) = \frac{1}{t}\sum_{\tau=1}^t \mathbb{1}\{b_{f_\tau}(\mathbf{z}, \mathbf{x}) = a\}$, *and* $\widehat{p}_1(\mathbb{1}_{(\sigma(\mathbf{z}, \mathbf{x}) = a)}) = \frac{1}{A_f}$ *for* $a_f \in \mathcal{A}_f$, *then the regret of Algorithm 1 satisfies* $\mathbb{E}[R(T)] = O(A_f\sqrt{T \log(T)})$.

### 4.2 Stochastic contexts and adversarial follower types

We now consider the setting in which the sequence of contexts are drawn i.i.d. from some unknown distribution $\mathcal{P}$ and the follower $f_t$ is chosen by an adversary with knowledge of $\mathcal{P}$ and $\mathbf{z}_1, \ldots, \mathbf{z}_{t-1}$,

---

[6]All that is required to run Algorithm 1 in a particular problem instance is an oracle for evaluating $\widehat{\mathbb{E}}_t[u(\mathbf{z}, \mathbf{x}, b_{f_t}(\mathbf{z}, \mathbf{x}))]$ and updating $\widehat{\mathbf{p}}_t$.

[7]In general $K$ has no dependence on $A_f$, and vice versa.

---

**Algorithm 2:** Learning with stochastic contexts: full feedback

---

**Input:** Set of weights $\Omega$

Let $\mathbf{q}_1[\pi^{(\boldsymbol{\omega})}] := 1$, $\mathbf{p}_1[\pi^{(\boldsymbol{\omega})}] := \frac{1}{|\Pi|}$ for all $\pi^{(\boldsymbol{\omega})} \in \Pi := \{\pi^{(\boldsymbol{\omega})}\}_{\boldsymbol{\omega} \in \Omega}$

**for** $t = 1, \ldots, T$ **do**

    Sample $\pi_t \sim \mathbf{p}_t$, $a_{l,t} \sim \pi_t(\mathbf{z}_t)$, receive utility $u(\mathbf{z}_t, a_{l,t}, b_{f_t}(\pi_t(\mathbf{z}_t)))$, observe type $f_t$.

    For each policy $\pi^{(\boldsymbol{\omega})} \in \Pi$, compute $\boldsymbol{\ell}_t[\pi^{(\boldsymbol{\omega})}] := -u(\mathbf{z}_t, \pi^{(\boldsymbol{\omega})}(\mathbf{z}_t), b_{f_t}(\mathbf{z}_t, \pi^{(\boldsymbol{\omega})}(\mathbf{z}_t)))$ and set

    $\mathbf{q}_{t+1}[\pi^{(\boldsymbol{\omega})}] = \exp(-\eta \sum_{s=1}^{t} \boldsymbol{\ell}_s[\pi^{(\boldsymbol{\omega})}])$, $\mathbf{p}_{t+1}[\pi^{(\boldsymbol{\omega})}] = \mathbf{q}_{t+1}[\pi^{(\boldsymbol{\omega})}] / \sum_{\pi^{(\boldsymbol{\omega}')} \in \Pi} \mathbf{q}_{t+1}[\pi^{(\boldsymbol{\omega}')}]$.

**end**

---

but not $\mathbf{z}_t$. As was the case in Section 4.1, we consider a relaxed notion of regret which compares the performance of the leader to the best policy in expectation, although now the expectation is taken with respect to the distribution over contexts $\mathcal{P}$.

**Definition 4.9** (Expected Contextual Stackelberg Regret, II)**.** *Given a distribution over contexts $\mathcal{P}$ and a sequence of followers $f_1, \ldots, f_T$, the leader's expected contextual Stackeleberg regret is*

$$\mathbb{E}[R(T)] := \mathbb{E}_{\mathbf{z}_1, \ldots, \mathbf{z}_T \sim \mathcal{P}} \left[ \sum_{t=1}^{T} u(\mathbf{z}_t, \pi^*(\mathbf{z}_t), b_{f_t}(\mathbf{z}_t, \pi^*(\mathbf{z}_t))) - u(\mathbf{z}_t, \mathbf{x}_t, b_{f_t}(\mathbf{z}_t, \mathbf{x}_t)) \right],$$

*where $\pi^*$ is the optimal policy given knowledge of $f_1, \ldots, f_T$ and $\mathcal{P}$.*

Our key insight is that when the sequence of contexts is generated stochastically, to obtain no-regret it suffices to (1) play a standard, off-the-shelf online learning algorithm (e.g. Hedge) over a finite (albeit exponentially-large) set of policies in order to find one which is approximately optimal and then (2) bound the resulting discretization error.

**Lemma 4.10.** *When the sequence of contexts is determined stochastically, the expected utility of any fixed policy $\pi$ may be written as*

$$\mathbb{E}_{\mathbf{z}_1, \ldots, \mathbf{z}_T} \left[ \sum_{t=1}^{T} u(\mathbf{z}_t, \pi(\mathbf{z}_t), b_{f_t}(\mathbf{z}_t, \pi(\mathbf{z}_t))) \right] = \sum_{i=1}^{K} \mathbb{E}_{\mathbf{z}}[u(\mathbf{z}, \pi(\mathbf{z}), b_{\alpha^{(i)}}(\mathbf{z}, \pi(\mathbf{z})))] \left( \sum_{t=1}^{T} \mathbb{E}_{\mathbf{z}_1, \ldots, \mathbf{z}_{t-1}}[\mathbb{1}\{f_t = \alpha^{(i)}\}] \right).$$

Using Lemma 4.10, we now show that it suffices to play Hedge over a finite set of policies $\Pi$ in order for the leader to obtain no-regret (Algorithm 2). The key step in our analysis is to show that the discretization error is small for our chosen policy class $\Pi$.[8] For a given weight vector $\boldsymbol{\omega} \in \mathbb{R}^K$, let $\pi^{(\boldsymbol{\omega})}(\mathbf{z}) := \arg\max_{\mathbf{z} \in \mathcal{E}_{\mathbf{z}}} \sum_{i=1}^{K} u(\mathbf{z}, \mathbf{x}, b_{\alpha^{(i)}}(\mathbf{z}, \mathbf{x})) \cdot \boldsymbol{\omega}[i]$. For a given set of weight vectors $\Omega$, we set $\Pi$ to be the induced policy class, i.e. $\Pi := \{\pi^{(\boldsymbol{\omega})}\}_{\boldsymbol{\omega} \in \Omega}$.

**Theorem 4.11.** *If $\Omega = \{\boldsymbol{\omega} \; : \; \boldsymbol{\omega} \in \Delta^K, T \cdot \boldsymbol{\omega}[i] \in \mathbb{N}, \forall i \in [K]\}$ and $\eta = \sqrt{\frac{\log \Pi}{T}}$, then Algorithm 2 obtains expected contextual Stackelberg regret (Definition 4.9) $\mathbb{E}[R(T)] = O\left(\sqrt{KT \log T} + K\right)$.*

We conclude by briefly comparing our results with those of the non-contextual Stackelberg game setting of Balcan et al. [5]. In particular, the setting of this subsection may be viewed as a generalization of the setting of Balcan et al. [5] in which the leader and follower utilities at time $t$ also depend on a stochastically-generated context $\mathbf{z}_t$. When $|\mathcal{Z}| = 1$, we recover their setting exactly. Under their non-contextual setting, there is only one set of approximate extreme points $\mathcal{E}_{\mathbf{z}}$, and so we write the $\mathcal{E} = \mathcal{E}_{\mathbf{z}}$. Here it suffices to consider the set of constant "policies" which always map to one of the (approximate) extreme points in $\mathcal{E}$. Plugging this choice of $\Pi$ into Algorithm 2, we recover their algorithm (and therefore also their regret rates) exactly.

However, it is also worth noting that more care must be taken to obtain regret guarantees against an adaptive adversary in our setting compared to the non-contextual setting of Balcan et al. [5].[9] In particular, we need to bound the discretization error due to considering a finite set of policies, but it is without loss of generality to consider a finite set of mixed strategies in the non-contextual setting.

---

[8]Interestingly, there is no discretization error if the sequence of follower types is chosen by a *non-adaptive* adversary. To see this, one can repeat the proof of Lemma 4.10, using the fact that $f_1, \ldots, f_T$ is independent from the realized draws $\mathbf{z}_1, \ldots, \mathbf{z}_T$ when the sequence of follower types is chosen by a non-adaptive adversary.

[9]Indeed, our notion of contextual Stackelberg regret is stronger than their non-contextual version of regret.

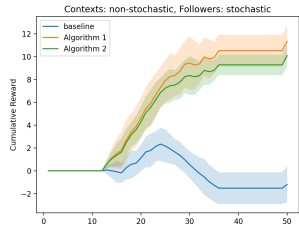
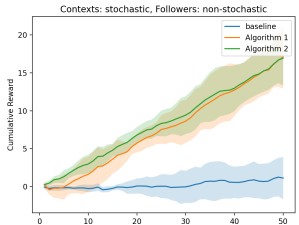
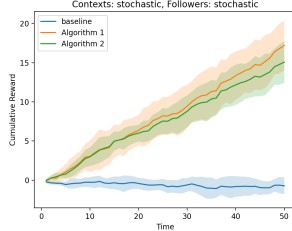

(a) Non-stochastic contexts, stochas-· tic follower types.    (b) Stochastic contexts, non-stochastic follower types.    (c) Stochastic contexts, stochastic follower types.

Figure 2: Cumulative average reward of Algorithm 1, Algorithm 2, and the algorithm of Balcan et al. [5] (which does not take side information into consideration) over five runs in a synthetic data setup. Shaded regions represent one standard deviation.

### 4.3   Simulations

We empirically evaluate the performance of Algorithm 1 and Algorithm 2 on synthetically-generated data. We consider a setup in which $K = 5$, $A = A_f = 3$, and the context dimension $d = 3$. Utility functions are linear in both the context and player actions, and are sampled u.a.r. from $[-1, 1]^{3 \times 3 \times 3}$.

We compare the cumulative reward of our algorithms to each other and the algorithm of Balcan et al. [5] (which does not leverage side information) as a baseline. We simulate non-stochastic context arrivals in Figure 2a by displaying the same context for $T/4$ time-steps in a row. Follower types are chosen u.a.r. from each of the five follower types. In Figure 2b, contexts are generated stochastically by sampling each component u.a.r. from $[-1, 1]$. Followers are chosen non-stochastically by deterministically cycling over the five types. In Figure 2c, both contexts and follower types are chosen stochastically. Specifically, contexts are generated as in Figure 2b and follower types are generated as in Figure 2a.

We find that Algorithm 1 and Algorithm 2 perform similarly across instances, and both significantly out-perform the baseline of Balcan et al. [5]. It would be interesting to find instances for which Algorithm 1 (resp. Algorithm 2) performs poorly whenever followers (resp. contexts) are chosen non-stochastically.

## 5   Extension to bandit feedback

We have so far assumed that the leader gets to observe the follower's type after each round. However this assumption may not always hold in real-world Stackelberg game settings. For example, in cyber security domains it may be hard to deduce the organization responsible for a failed cyber attack. In wildlife protection, a very successful poacher may never be seen by the park rangers. Instead, the leader may only be able to observe the action the follower takes at each round. Following previous work on learning in non-contextual Stackelberg games, we refer to this type of feedback as *bandit* feedback. What can we say about the leader's ability to learn under bandit feedback when there is side information?

While our impossibility result of Section 3 immediately applies to this more challenging setting, our algorithms from Section 4 do not. This is because we can no longer compute quantities such as $\mathbb{1}\{f_t = \alpha^{(i)}\}$ or $b_{f_t}(\mathbf{z}_t, \mathbf{x})$ for an arbitrary mixed strategy $\mathbf{x}$ from just follower $f_t$'s action alone. We still assume that the follower is one of $K$ different types, although $f_t$ is now *never* revealed to the leader.

We allow ourselves two relaxations when designing learning algorithms which operate under bandit feedback. First, while the leader's utility function may still depend on the context $\mathbf{z}_t$, we assume that the follower's utility is a function of the leader's mixed strategy $\mathbf{x}_t$ alone, i.e. $u_f(\mathbf{z}, \mathbf{x}, a_f) = u_f(\mathbf{x}, a_f)$ for all $\mathbf{z} \in \bar{\mathcal{Z}}$. This allows us to drop the dependence on $\mathbf{z}_t$ from both the follower's best response and the set of approximate extreme points, i.e. $b_f(\mathbf{z}, \mathbf{x})$ becomes $b_f(\mathbf{x})$ and $\mathcal{E}_\mathbf{z}$ becomes $\mathcal{E}$. Furthermore, our definitions of contextual follower best-response region (Definition 4.1) and contextual best-response region (Definition 4.2) collapse to their non-contextual counterparts. Depending on the application domain, the assumption that only the leader's utility depends on the side

information may be reasonable. For instance, while an institution would prefer that a server with less traffic is hacked compared to one with more, a hacker might only care about the information hosted on the server (which may not be related to network traffic patterns). Second, we design algorithms with regret guarantees which only hold against a *non-adaptive* adversary.[10] Despite these relaxations, the problem of learning under bandit feedback still remains challenging because of the exponentially large size of $\mathcal{E}$. While a natural first step is to estimate $p(b_{f_t}(\mathbf{x}) = a_f)$ (i.e. the probability that follower at round $t$ best-responds with action $a_f$ when the leader plays mixed strategy $\mathbf{x}$) for all $\mathbf{x} \in \mathcal{E}$ and $a_f \in \mathcal{A}_f$, doing so naively would take exponentially-many rounds, due to the size of $\mathcal{E}$.

Building off of results in the non-contextual setting of Balcan et al. [5], we leverage the fact that the leader's utility for different mixed strategies is not independent. Instead, they are *linearly* related through the frequency of follower types which take a particular action, given a particular leader mixed strategy. Therefore, it suffices to estimate this linear function (which can be done using as few as $K$ samples) to get an unbiased estimate of $p(b_{f_t}(\mathbf{x}) = a_f)$ for any $\mathbf{x} \in \mathcal{E}$ and $a_f \in \mathcal{A}_f$. Borrowing from the literature on linear bandits, we use a *barycentric spanner* [4] to estimate $\{\{p(b_{f_t}(\mathbf{x}) = a_f)\}_{a_f \in \mathcal{A}_f}\}_{\mathbf{x} \in \mathcal{E}}$ in both partial adversarial settings we consider. A barycentric spanner for compact vector space $\mathcal{W}$ is a special basis such that any vector in $\mathcal{W}$ may be expressed as a linear combination of elements in the basis, *with each linear coefficient being in the range* $[-1, 1]$.

In Appendix C.1, we use the property that estimators constructed using barycentric spanners have *low variance* to show that an explore-then-exploit algorithm achieves $O(K^{2/3} A_f^{2/3} T^{2/3} \log^{1/3} T)$ expected contextual Stackelberg regret in the setting with stochastic follower types and adversarial contexts. Specifically, our algorithm (Algorithm 3) plays a special set of $K$ mixed strategies $N$ times each, then uses barycentric spanners to estimate $\{p_t(\mathbb{1}_{(\sigma(\mathbf{z}, \mathbf{x}) = a_f)})\}_{a_f \in \mathcal{A}_f}$ for all $\mathbf{x} \in \mathcal{X}$ and $\mathbf{z} \in \mathcal{Z}$, after which Algorithm 3 plays greedily like in Section 4.1.

In Appendix C.2, we use the property that estimators constructed using barycentric spanners are *bounded* to design a reduction to our algorithm in Section 4.2 which achieves $O(K A_f^{1/3} T^{2/3} \log^{1/3} T)$ expected contextual Stackelberg regret whenever the sequence of contexts is chosen stochastically and the sequence of follower types is chosen by an adversary. Finally, while it may be possible to obtain $O(\sqrt{T})$ regret without using barycentric spanners, this would come at the cost of a linear dependence on $|\mathcal{E}|$ (and therefore an *exponential* dependence on $K$ and $A_f$) in regret.

## 6 Conclusion

We initiate the study of Stackelberg games with side information, which despite the presence of side information in many Stackelberg game settings, has not received attention from the community. We focus on the online setting in which the leader faces a sequence of contexts and follower types. We show that when both sequences are chosen adversarially, no-regret learning is not possible even for highly structured policy classes. When either sequence is chosen stochastically, we obtain algorithms with $\tilde{O}(\sqrt{T})$ regret. We also explore an extension to bandit feedback, in which we obtain $\tilde{O}(T^{2/3})$ regret in both settings. There are several exciting avenues for future research; we highlight two below.

1. **Intermediate forms of adversary.** The two relaxations of the fully adversarial setting that we consider, while natural, rule out the leader learning about the follower's type from the context. Although we prove that learning is impossible in the fully adversarial setting, our lower bound does not rule out, e.g. settings where the mapping from contexts to follower types has finite Littlestone dimension. It would be interesting to further explore this direction to pin down when no-regret learning is possible.

2. $\tilde{O}(T^{1/2})$ **regret under bandit feedback.** Bernasconi et al. [6] obtain $O(T^{1/2})$ regret when learning in non-contextual Stackelberg games under bandit feedback against an adversarially-chosen sequence of follower types via a reduction to adversarial linear bandits. However, applying similar steps to Bernasconi et al. in our setting results in a reduction to a generalization of the (adversarial) contextual bandit problem for which we are not aware of any regret minimizing algorithm. Nevertheless, we view exploring whether $\tilde{O}(T^{1/2})$ contextual Stackelberg regret is possible under bandit feedback as a natural and exciting future direction.

---

[10]We hypothesize that our results in this section could be extended to hold against an adaptive adversary by using more clever exploration strategies.

## Acknowledgements

This work was supported in part by NSF Grants CCF-1910321, #1763786, and by an NDSEG fellowship. The authors would like to thank the anonymous reviewers for valuable feedback, and Martino Bernasconi, Matteo Castiglioni, and Andrea Celli for helpful discussions surrounding related work.

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

# A   Appendix for Section 3: On the impossibility of fully adversarial no-regret learning

**Theorem 3.2.** *If an adversary can choose both the sequence of contexts $\mathbf{z}_1, \ldots, \mathbf{z}_T$ and the sequence of followers $f_1, \ldots, f_T$, no algorithm can achieve better than $\Omega(T)$ contextual Stackelberg regret in expectation over the internal randomness of the algorithm, even when $\pi^*$ is restricted to come from the set of linear thresholding functions.*

*Proof.* We proceed via proof by contradiction. Assume that there exists an algorithm ALG which achieves $o(T)$ contextual Stackelberg regret against an adversarially-chosen sequence of contexts and follower types. Note that at every time-step, ALG takes as input a context $\mathbf{z}_t$ and produces a mixed strategy $\mathbf{x}_t$.

We now describe the family of contextual Stackelberg game instances we reduce to. Consider the setting in which there are two follower types ($\alpha^{(1)}$ and $\alpha^{(2)}$) and two leader/follower actions ($\mathcal{A} = \mathcal{A}_f = \{a_1, a_2\}$). Suppose that the context space is of the form $\mathcal{Z} = [0, 1]$, and that regardless of the realized context or leader mixed strategy, the best-response of follower type $\alpha^{(1)}$ is to play action $a_1$ ($b_{\alpha^{(1)}}(\mathbf{z}, \mathbf{x}) = a_1, \forall \mathbf{z} \in \mathcal{Z}, \mathbf{x} \in \mathcal{X}$) and the best-response of follower type $\alpha^{(2)}$ is to play action $a_2$ ($b_{\alpha^{(2)}}(\mathbf{z}, \mathbf{x}) = a_2, \forall \mathbf{z} \in \mathcal{Z}, \mathbf{x} \in \mathcal{X}$). Since the follower's best-response does not depend on the leader's mixed strategy or the context, we use the shorthand $b_{f_t} := b_{f_t}(\mathbf{z}_t, \mathbf{x}_t)$. Finally, suppose that the leader's utility function is given by $u(\mathbf{z}, a_l, a_f) = \mathbb{1}\{a_l = a_f\}$. Note that this is a special case of our general setting (described in Section 2).

The reduction from online linear thresholding proceeds as follows. In each round $t \in [T]$,

1. Given a point $\omega_t \in [0, 1]$, we give the context $z_t = \omega_t$ as input to ALG.

2. In return, we receive mixed strategy $\mathbf{x}_t \in \Delta(\{a_1, a_2\})$ from ALG. We set $g_t = \mathbf{x}_t[1]$.[11]

3. Play according to $g_t$, and receive label $y_t$ and utility $u_{\mathrm{OLT}}(\omega_t, g_t)$ from Nature. Give follower type
$$f_t = \begin{cases} \alpha^{(1)} & \text{if } y_t = 1 \\ \alpha^{(2)} & \text{if } y_t = -1 \end{cases}$$
and utility $u(\mathbf{z}_t, \mathbf{x}_t, b_{f_t}) = \mathbf{x}_t[1] \cdot \mathbb{1}\{b_{f_t} = a_1\} + \mathbf{x}_t[2] \cdot \mathbb{1}\{b_{f_t} = a_2\}$ as input to ALG.

Observe that under this reduction,
$$\pi^*(\mathbf{z}) = \begin{cases} [1 \ 0]^\top & \text{if } z > s \text{ and} \\ [0 \ 1]^\top & \text{otherwise.} \end{cases} \tag{2}$$

since if $z > s$, $f_t = \alpha^{(1)}$ and otherwise $f_t = \alpha^{(0)}$. By playing according to $\pi^*$, we can ensure that $u(z_t, \pi^*(z_t), b_{f_t}) = 1$ for all $t \in [T]$. $\pi^*$ must then be optimal, because 1 is the largest possible per-round utility that the leader can receive.

Since ALG achieves no-contextual-Stackelberg-regret, we know by Definition 2.3 that
$$R(T) = \sum_{t=1}^{T} u(\mathbf{z}_t, \pi^*(\mathbf{z}_t), b_{f_t}) - u(\mathbf{z}_t, \mathbf{x}_t, b_{f_t}) = o(T). \tag{3}$$

To conclude, it suffices to show that $R_{\mathrm{OLT}}(T) = o(T)$ using Equation (2) and Equation (3). Applying Equation (2), we see that
$$R(T) = T - \sum_{t=1}^{T}(\mathbf{x}_t[1] \cdot \mathbb{1}\{b_{f_t} = a_1\} + \mathbf{x}_t[2] \cdot \mathbb{1}\{b_{f_t} = a_2\}). \tag{4}$$

By construction, $\mathbb{1}\{b_{f_t} = a_1\} = \mathbb{1}\{y_t = 1\}$, $\mathbb{1}\{b_{f_t} = a_2\} = \mathbb{1}\{y_t = -1\}$, $\mathbf{x}_t[1] = g_t$, and $\mathbf{x}_t[2] = 1 - g_t$. Substituting this into Equation (4), we see that
$$R(T) = T - \sum_{t=1}^{T}(g_t \cdot \mathbb{1}\{y_t = 1\} + (1 - g_t) \cdot \mathbb{1}\{y_t = -1\}) =: R_{\mathrm{OLT}}(T). \tag{5}$$

---

[11]Observe that $\mathbf{x}_t[2] = 1 - g_t$.

By Equation (3) and Equation (5), we can conclude that $R_{\mathrm{OLT}}(T) = o(T)$. However, this is a contradiction since no no-regret learning algorithm exists for the online linear thresholding problem by Lemma 3.1. Therefore it must not be possible to achieve no-contextual-Stackelberg-regret whenever the sequence of contexts and follower types is chosen by an adversary.

$\square$

# B  Appendix for Section 4: Limiting the power of the adversary

**Lemma 4.4.** *For any sequence of followers $f_1, \ldots f_T$ and any leader policy $\pi$, there exists a policy $\pi^{(\mathcal{E})} : \mathcal{Z} \to \cup_{\mathbf{z} \in \mathcal{Z}} \mathcal{E}_{\mathbf{z}}$ that, when given context $\mathbf{z}$, plays a mixed strategy in $\mathcal{E}_{\mathbf{z}}$ and guarantees that $\sum_{t=1}^{T} u(\mathbf{z}_t, \pi(\mathbf{z}_t), b_{f_t}(\mathbf{z}_t, \pi(\mathbf{z}_t))) - u(\mathbf{z}_t, \pi^{(\mathcal{E})}(\mathbf{z}_t), b_{f_t}(\mathbf{z}_t, \pi^{(\mathcal{E})}(\mathbf{z}_t))) \leq 1$. Moreover, the same result holds in expectation over any distribution over follower types $\mathcal{F}$.*

*Proof.* Observe that for any $\mathbf{z}$,

$$\pi^*(\mathbf{z}) := \arg\max_{\mathbf{x} \in \Delta(\mathcal{A}_l)} \sum_{t=1}^{T} \mathbb{1}\{\mathbf{z}_t = \mathbf{z}\} \sum_{a_l \in \mathcal{A}_l} \mathbf{x}[a_l] \cdot u(\mathbf{z}, a_l, b_{f_t}(\mathbf{z}, \mathbf{x}))$$

$$= \arg\max_{\mathbf{x} \in \Delta(\mathcal{A}_l)} \sum_{i=1}^{K} \sum_{a_l \in \mathcal{A}_l} \mathbf{x}[a_l] \cdot u(\mathbf{z}, a_l, b_{\alpha^{(i)}}(\mathbf{z}, \mathbf{x})) \sum_{t=1}^{T} \mathbb{1}\{\mathbf{z}_t = \mathbf{z}, f_t = \alpha^{(i)}\}$$

The solution to the above optimization may be obtained by first solving

$$\mathbf{x}_{a_{1:K}}(\mathbf{z}) = \arg\max_{\mathbf{x} \in \Delta(\mathcal{A}_l)} \sum_{i=1}^{K} \sum_{a_l \in \mathcal{A}_l} \mathbf{x}[a_l] \cdot u(\mathbf{z}, a_l, a^{(i)}) \cdot \sum_{t=1}^{T} \mathbb{1}\{\mathbf{z}_t = \mathbf{z}, f_t = \alpha^{(i)}\}$$

s.t. $b_{\alpha^{(i)}}(\mathbf{z}, \mathbf{x}) = a^{(i)}, \forall i \in [K]$

(6)

for every possible setting of $a^{(1)}, \ldots, a^{(K)}$, and then taking the maximum of all feasible solutions. Since Equation (6) is an optimization over contextual best-response region $\mathcal{X}_{\mathbf{z}}(a^{(1)}, \ldots, a^{(K)})$ and all contextual best-response regions are convex polytopes, $\pi^*(\mathbf{z})$ will be an extreme point of some contextual best-response region, although it may not be attained. Overloading notation, let $\mathcal{X}_{\mathbf{z}}(\pi^*(\mathbf{z}))$ denote the contextual best-response region corresponding to $\pi^*(\mathbf{z})$, i.e., $\pi^*(\mathbf{z}) \in \mathcal{X}_{\mathbf{z}}(\pi^*(\mathbf{z}))$. Since for a fixed context $\mathbf{z} \in \mathcal{Z}$ the leader's utility is a linear function of $\mathbf{x}$ over the convex polytope $\mathcal{X}_{\mathbf{z}}(\pi^*(\mathbf{z}))$, there exists a point $\mathbf{x}(\mathbf{z}) \in \mathrm{cl}(\mathcal{X}_{\mathbf{z}}(\pi^*(\mathbf{z})))$ such that

$$\sum_{t=1}^{T} u(\mathbf{z}, \mathbf{x}(\mathbf{z}), b_{f_t}(\mathbf{z}, \pi^*(\mathbf{z}))) \cdot \mathbb{1}\{\mathbf{z}_t = \mathbf{z}\} \geq \sum_{t=1}^{T} u(\mathbf{z}, \pi^*(\mathbf{z}), b_{f_t}(\mathbf{z}, \pi^*(\mathbf{z}))) \cdot \mathbb{1}\{\mathbf{z}_t = \mathbf{z}\}.$$

Let $\mathbf{x}'(\mathbf{z})$ denote the corresponding point in $\mathcal{E}_{\mathbf{z}}$ such that $\|\mathbf{x}'(\mathbf{z}) - \mathbf{x}(\mathbf{z})\|_1 \leq \delta$. (Such a point will always exist by Definition 4.3.) Since $u \in [0, 1]$ and is linear in $\mathbf{x}$ for a fixed context and follower best-response,

$$\sum_{t=1}^{T} u(\mathbf{z}, \mathbf{x}'(\mathbf{z}), b_{f_t}(\mathbf{z}, \mathbf{x}'(\mathbf{z}))) \cdot \mathbb{1}\{\mathbf{z}_t = \mathbf{z}\} = \sum_{t=1}^{T} u(\mathbf{z}, \mathbf{x}'(\mathbf{z}), b_{f_t}(\mathbf{z}, \pi^*(\mathbf{z}))) \cdot \mathbb{1}\{\mathbf{z}_t = \mathbf{z}\}$$

$$\geq \sum_{t=1}^{T} (u(\mathbf{z}, \mathbf{x}(\mathbf{z}), b_{f_t}(\mathbf{z}, \pi^*(\mathbf{z}))) - \delta) \cdot \mathbb{1}\{\mathbf{z}_t = \mathbf{z}\}$$

$$\geq \sum_{t=1}^{T} (u(\mathbf{z}, \pi^*(\mathbf{z}), b_{f_t}(\mathbf{z}, \pi^*(\mathbf{z}))) - \delta) \cdot \mathbb{1}\{\mathbf{z}_t = \mathbf{z}\}$$

Summing over all unique $\mathbf{z}$ encountered by the algorithm over $T$ time-steps, we obtain the desired result for the policy $\pi^{(\mathcal{E})}$ which plays mixed strategy $\pi^{(\mathcal{E})}(\mathbf{z}) = \mathbf{x}'(\mathbf{z})$ when given context $\mathbf{z}$. Finally, observe that the same line of reasoning holds whenever we are interested in the optimal policy *in expectation with respect to some distribution $\mathcal{F}$ over followers*, as is the case in, e.g. Section 4.1 (with $\sum_{t=1}^{T} \mathbb{1}\{\mathbf{z}_t = \mathbf{z}, f_t = \alpha^{(i)}\}$ replaced with $\mathbb{P}(f = \alpha^{(i)})$). $\square$

## B.1 Section 4.1: Stochastic follower types and adversarial contexts

**Theorem 4.6.** *Let* $\mathbf{p}(\mathbf{z}, \mathbf{x}) := [p(b_{f_t}(\mathbf{z}, \mathbf{x}) = a_f)]_{a_f \in \mathcal{A}_f}$ *and* $\widehat{\mathbf{p}}_t(\mathbf{z}, \mathbf{x}) := [\widehat{p}_t(b_{f_t}(\mathbf{z}, \mathbf{x}) = a_f)]_{a_f \in \mathcal{A}_f}$. *The expected contextual Stackelberg regret (Definition 4.5) of Algorithm 1 satisfies*

$$\mathbb{E}[R(T)] \leq 1 + 2\sum_{t=1}^{T} \mathbb{E}_{f_1,\ldots,f_{t-1}}[\text{TV}(\mathbf{p}(\mathbf{z}_t, \pi^{(\mathcal{E})}(\mathbf{z}_t)), \widehat{\mathbf{p}}_t(\mathbf{z}_t, \pi^{(\mathcal{E})}(\mathbf{z}_t))) + \text{TV}(\mathbf{p}(\mathbf{z}_t, \pi_t(\mathbf{z}_t)), \widehat{\mathbf{p}}_t(\mathbf{z}_t, \pi_t(\mathbf{z}_t)))].$$

*Proof of Theorem 4.6.* For any $\mathbf{z} \in \mathcal{Z}$ and $t \in [T]$,

$$\widehat{\mathbb{E}}_t[u(\mathbf{z}, \pi^{(\mathcal{E})}(\mathbf{z}), b_f(\mathbf{z}, \pi^{(\mathcal{E})}(\mathbf{z})))] \leq \mathbb{E}_t[u(\mathbf{z}, \pi_t(\mathbf{z}), b_f(\mathbf{z}, \pi_t(\mathbf{z})))]$$
$$+ \widehat{\mathbb{E}}_t[u(\mathbf{z}, \pi_t(\mathbf{z}), b_f(\mathbf{z}, \pi_t(\mathbf{z})))] - \mathbb{E}_t[u(\mathbf{z}, \pi_t(\mathbf{z}), b_f(\mathbf{z}, \pi_t(\mathbf{z})))].$$

Since utilities are bounded in $[0, 1]$ and the expectations $\mathbb{E}_t$ and $\widehat{\mathbb{E}}_t$ are taken with respect to $\mathbf{p}$ and $\widehat{\mathbf{p}}_t$ respectively, we can upper-bound $\widehat{\mathbb{E}}_t[u(\mathbf{z}, \pi_t(\mathbf{z}), b_f(\mathbf{z}, \pi_t(\mathbf{z})))] - \mathbb{E}_t[u(\mathbf{z}, \pi_t(\mathbf{z}), b_f(\mathbf{z}, \pi_t(\mathbf{z})))]$ by

$$\int |d\widehat{p}_t(\mathbf{z}, \pi_t(\mathbf{z})) - dp_t(\mathbf{z}, \pi_t(\mathbf{z}))| = 2\text{TV}(\mathbf{p}(\mathbf{z}, \pi_t(\mathbf{z})), \widehat{\mathbf{p}}_t(\mathbf{z}, \pi_t(\mathbf{z}))).$$

Putting everything together, we get that

$$\widehat{\mathbb{E}}_t[u(\mathbf{z}, \pi^{(\mathcal{E})}(\mathbf{z}), b_f(\mathbf{z}, \pi^{(\mathcal{E})}(\mathbf{z})))] \leq \mathbb{E}_t[u(\mathbf{z}, \pi_t(\mathbf{z}), b_f(\mathbf{z}, \pi_t(\mathbf{z})))] + 2\text{TV}(\mathbf{p}(\mathbf{z}, \pi_t(\mathbf{z})), \widehat{\mathbf{p}}_t(\mathbf{z}, \pi_t(\mathbf{z}))).$$

We now use this fact to bound the expected regret. By Lemma 4.4,

$$\mathbb{E}[R(T)] \leq 1 + \sum_{t=1}^{T} \mathbb{E}_{f_1,\ldots,f_t}[u(\mathbf{z}_t, \pi^{(\mathcal{E})}(\mathbf{z}_t), b_{f_t}(\mathbf{z}_t, \pi^{(\mathcal{E})}(\mathbf{z}_t))) - u(\mathbf{z}_t, \pi_t(\mathbf{z}_t), b_{f_t}(\mathbf{z}_t, \pi_t(\mathbf{z}_t)))]$$

$$\leq 1 + \sum_{t=1}^{T} \mathbb{E}_{f_1,\ldots,f_{t-1}}[\mathbb{E}_t[u(\mathbf{z}_t, \pi^{(\mathcal{E})}(\mathbf{z}_t), b_{f_t}(\mathbf{z}_t, \pi^{(\mathcal{E})}(\mathbf{z}_t)))]$$

$$- \widehat{\mathbb{E}}_t[u(\mathbf{z}, \pi^{(\mathcal{E})}(\mathbf{z}), b_{f_t}(\mathbf{z}, \pi^{(\mathcal{E})}(\mathbf{z})))] + 2\text{TV}(\mathbf{p}(\mathbf{z}, \pi_t(\mathbf{z})), \widehat{\mathbf{p}}_t(\mathbf{z}, \pi_t(\mathbf{z})))].$$

By repeating the same steps as above, we can upper-bound

$$\mathbb{E}_t[u(\mathbf{z}_t, \pi^{(\mathcal{E})}(\mathbf{z}_t), b_f(\mathbf{z}_t, \pi^{(\mathcal{E})}(\mathbf{z}_t)))] - \widehat{\mathbb{E}}_t[u(\mathbf{z}, \pi^{(\mathcal{E})}(\mathbf{z}), b_f(\mathbf{z}, \pi^{(\mathcal{E})}(\mathbf{z})))]$$

by $2\text{TV}(\mathbf{p}(\mathbf{z}_t, \pi^{(\mathcal{E})}), \widehat{\mathbf{p}}_t(\mathbf{z}_t, \pi^{(\mathcal{E})}))$. This gets us the desired result. $\square$

**Corollary 4.7.** *If* $\widehat{\mathbf{p}}_t = \{\widehat{p}_t(f_t = \alpha^{(i)})\}_{i \in [K]}$, $\widehat{p}_{t+1}(f = \alpha^{(i)}) = \frac{1}{t}\sum_{\tau=1}^{t} \mathbb{1}\{f_\tau = \alpha^{(i)}\}$, *and* $\widehat{p}_1(f = \alpha^{(i)}) = \frac{1}{K}$ *for* $i \in [K]$, *then the regret of Algorithm 1 satisfies* $\mathbb{E}[R(T)] = O(K\sqrt{T \log(T)})$.

*Proof.* For $t \geq 2$,

$$\text{TV}(\mathbf{p}(\mathbf{z}, \mathbf{x}), \widehat{\mathbf{p}}_t(\mathbf{z}, \mathbf{x})) = \frac{1}{2}\sum_{i=1}^{K} |p_t(f_t = \alpha^{(i)}) - \widehat{p}_t(f_t = \alpha^{(i)})|$$

$$= \frac{1}{2}\sum_{i=1}^{K} \frac{1}{t-1}\left|\sum_{\tau=1}^{t-1} \mathbb{1}\{f_\tau = \alpha^{(i)}\} - \mathbb{E}_{f_\tau}[\mathbb{1}\{f_\tau = \alpha^{(i)}\}]\right|$$

for any $\mathbf{z} \in \mathcal{Z}$ and $\mathbf{x} \in \mathcal{X}$. By Hoeffding's inequality, we know that

$$\frac{1}{t-1}\left|\sum_{\tau=1}^{t-1} \mathbb{1}\{f_{\tau=\alpha^{(i)}}\} - \mathbb{E}_{f_\tau}[\mathbb{1}\{f_{\tau=\alpha^{(i)}}\}]\right| \leq 2\sqrt{\frac{\log(2T)}{t-1}}$$

simultaneously for all $t \in [T]$ and $i \in [K]$, with probability at least $1 - \frac{1}{T^2}$. Dropping the dependence of $\mathbf{p}, \widehat{\mathbf{p}}_t$ on $\mathbf{z}$ and $\mathbf{x}$, we can conclude that

$$\mathbb{E}_{f_1,\ldots,f_{t-1}}[\text{TV}(\mathbf{p}, \widehat{\mathbf{p}}_t)] \leq K\sqrt{\frac{\log(2T)}{t-1}} + \frac{1}{2T}$$

(since $K \leq T$), and so

$$\mathbb{E}[R(T)] \leq 1 + 4 \sum_{t=1}^{T} K \left( \sqrt{\frac{\log(2T)}{t-1}} + \frac{1}{2T} \right) = O\left( K \sqrt{T \log(T)} \right).$$

$\square$

**Corollary 4.8.** *If* $\widehat{\mathbf{p}}_t(\mathbf{z}, \mathbf{x}) = \{\widehat{p}_t(\mathbb{1}_{(\sigma(\mathbf{z}, \mathbf{x}) = a_f)})\}_{a_f \in \mathcal{A}_f}$, $\widehat{p}_{t+1}(\mathbb{1}_{(\sigma(\mathbf{z}, \mathbf{x}) = a)}) = \frac{1}{t} \sum_{\tau=1}^{t} \mathbb{1}\{b_{f_\tau}(\mathbf{z}, \mathbf{x}) = a\}$, *and* $\widehat{p}_1(\mathbb{1}_{(\sigma(\mathbf{z}, \mathbf{x}) = a)}) = \frac{1}{A_f}$ *for* $a_f \in \mathcal{A}_f$, *then the regret of Algorithm 1 satisfies* $\mathbb{E}[R(T)] = O(A_f \sqrt{T \log(T)})$.

*Proof.* For $t \geq 2$,

$$\mathrm{TV}(\mathbf{p}(\mathbf{z}, \mathbf{x}), \widehat{\mathbf{p}}_t(\mathbf{z}, \mathbf{x})) = \frac{1}{2} \sum_{a \in \mathcal{A}_f} |p_t(\mathbb{1}_{(\sigma(\mathbf{z}, \mathbf{x}) = a)}) - \widehat{p}_t(\mathbb{1}_{(\sigma(\mathbf{z}, \mathbf{x}) = a)})|$$

$$= \frac{1}{2} \sum_{a \in \mathcal{A}_f} |p(\mathbb{1}_{(\sigma(\mathbf{z}, \mathbf{x}) = a)}) - \frac{1}{t-1} \sum_{\tau=1}^{t-1} \mathbb{1}\{b_{f_\tau}(\mathbf{z}, \mathbf{x}) = a\}|$$

$$= \frac{1}{2} \sum_{a \in \mathcal{A}_f} \frac{1}{t-1} \left| \sum_{\tau=1}^{t-1} \mathbb{1}\{b_{f_\tau}(\mathbf{z}, \mathbf{x}) = a\} - \mathbb{E}_{f_\tau}[\mathbb{1}\{b_{f_\tau}(\mathbf{z}, \mathbf{x}) = a\}] \right|$$

for any $\mathbf{z} \in \mathcal{Z}$, $\mathbf{x} \in \mathcal{X}$. By Hoeffding's inequality,

$$\frac{1}{t-1} \left| \sum_{\tau=1}^{t-1} \mathbb{1}\{b_{f_\tau}(\mathbf{z}, \mathbf{x}) = a\} - \mathbb{E}_{f_\tau}[\mathbb{1}\{b_{f_\tau}(\mathbf{z}, \mathbf{x}) = a\}] \right| \leq 2\sqrt{\frac{\log(2T)}{t-1}}$$

simultaneously for all $t \in [T]$ and $i \in [K]$, with probability at least $1 - \frac{1}{T^2}$. Using this fact, we can conclude that

$$\mathbb{E}_{f_1, \ldots, f_{t-1}}[\mathrm{TV}(\mathbf{p}(\mathbf{z}, \mathbf{x}), \widehat{\mathbf{p}}_t(\mathbf{z}, \mathbf{x}))] \leq A_f \sqrt{\frac{\log(2T)}{t-1}} + \frac{1}{2T}$$

(since $K \leq T$) and

$$\mathbb{E}[R(T)] \leq 1 + 4 \sum_{t=1}^{T} \left( A_f \sqrt{\frac{\log(2T)}{t-1}} + \frac{1}{2T} \right)$$

$$\leq 3 + 4 A_f \sqrt{\log(2T)} \int_{t=0}^{T} \frac{1}{\sqrt{t}} dt$$

$$= O\left( A_f \sqrt{T \log(T)} \right).$$

$\square$

## B.2 Section 4.2: Stochastic contexts and adversarial follower types

The following regret guarantee for Hedge is a well-known result. (See, e.g. Gupta [11].)

**Lemma B.1.** *Hedge enjoys expected regret rate* $O(\sqrt{T \log n})$ *when there are $n$ actions, the learning rate is chosen to be* $\eta = \sqrt{\frac{\log n}{T}}$, *and the sequence of utilities for each arm are chosen by an adversary.*

**Lemma 4.10.** *When the sequence of contexts is determined stochastically, the expected utility of any fixed policy $\pi$ may be written as*

$$\mathbb{E}_{\mathbf{z}_1, \ldots, \mathbf{z}_T}\left[ \sum_{t=1}^{T} u(\mathbf{z}_t, \pi(\mathbf{z}_t), b_{f_t}(\mathbf{z}_t, \pi(\mathbf{z}_t))) \right] = \sum_{i=1}^{K} \mathbb{E}_{\mathbf{z}}[u(\mathbf{z}, \pi(\mathbf{z}), b_{\alpha^{(i)}}(\mathbf{z}, \pi(\mathbf{z})))] \left( \sum_{t=1}^{T} \mathbb{E}_{\mathbf{z}_1, \ldots, \mathbf{z}_{t-1}}[\mathbb{1}\{f_t = \alpha^{(i)}\}] \right).$$

*Proof.* For any fixed policy $\pi$,

$$\mathbb{E}_{\mathbf{z}_1,\ldots,\mathbf{z}_T}\left[\sum_{t=1}^{T} u(\mathbf{z}_t, \pi(\mathbf{z}_t), b_{f_t}(\mathbf{z}_t, \pi(\mathbf{z}_t)))\right] = \sum_{t=1}^{T} \mathbb{E}_{\mathbf{z}_1,\ldots,\mathbf{z}_t}\left[\sum_{i=1}^{K} u(\mathbf{z}_t, \pi(\mathbf{z}_t), b_{\alpha^{(i)}}(\mathbf{z}_t, \pi(\mathbf{z}_t)))\mathbb{1}\{f_t = \alpha^{(i)}\}\right]$$

$$= \sum_{t=1}^{T}\sum_{i=1}^{K} \mathbb{E}_{\mathbf{z}_t}[u(\mathbf{z}_t, \pi(\mathbf{z}_t), b_{\alpha^{(i)}}(\mathbf{z}_t, \pi(\mathbf{z}_t)))]\mathbb{E}_{\mathbf{z}_1,\ldots,\mathbf{z}_{t-1}}[\mathbb{1}\{f_t = \alpha^{(i)}\}]$$

where the second line uses the fact that $f_t$ cannot depend on $\mathbf{z}_t$, and the result follows from the fact that $\mathbf{z}_1, \ldots, \mathbf{z}_T$ are i.i.d. $\qquad\square$

**Theorem 4.11.** *If $\Omega = \{\boldsymbol{\omega} \ : \ \boldsymbol{\omega} \in \Delta^K, T \cdot \boldsymbol{\omega}[i] \in \mathbb{N}, \forall i \in [K]\}$ and $\eta = \sqrt{\frac{\log \Pi}{T}}$, then Algorithm 2 obtains expected contextual Stackelberg regret (Definition 4.9) $\mathbb{E}[R(T)] = O\left(\sqrt{KT\log T} + K\right)$.*

*Proof.* By Lemma 4.4,

$$\mathbb{E}[R(T)] = \mathbb{E}_{\mathbf{z}_1,\ldots,\mathbf{z}_T}\left[\sum_{t=1}^{T} u(\mathbf{z}_t, \pi^*(\mathbf{z}_t), b_{f_t}(\mathbf{z}_t, \pi^*(\mathbf{z}_t))) - u(\mathbf{z}_t, \pi_t(\mathbf{z}_t), b_{f_t}(\mathbf{z}_t, \pi_t(\mathbf{z}_t)))\right]$$

$$\leq \mathbb{E}_{\mathbf{z}_1,\ldots,\mathbf{z}_T}\left[\sum_{t=1}^{T} u(\mathbf{z}_t, \pi^{(\mathcal{E})}(\mathbf{z}_t), b_{f_t}(\mathbf{z}_t, \pi^{(\mathcal{E})}(\mathbf{z}_t))) - u(\mathbf{z}_t, \pi_t(\mathbf{z}_t), b_{f_t}(\mathbf{z}_t, \pi_t(\mathbf{z}_t)))\right] + 1$$

Let $\pi^{(\boldsymbol{\omega}^*)}$ denote the optimal-in-hindsight policy in $\Pi$.

$$\mathbb{E}[R(T)] \leq \mathbb{E}_{\mathbf{z}_1,\ldots,\mathbf{z}_T}\left[\sum_{t=1}^{T} u(\mathbf{z}_t, \pi^{(\boldsymbol{\omega}^*)}(\mathbf{z}_t), b_{f_t}(\mathbf{z}_t, \pi^{(\boldsymbol{\omega}^*)}(\mathbf{z}_t))) - u(\mathbf{z}_t, \pi_t(\mathbf{z}_t), b_{f_t}(\mathbf{z}_t, \pi_t(\mathbf{z}_t)))\right]$$

$$+ \mathbb{E}_{\mathbf{z}_1,\ldots,\mathbf{z}_T}\left[\sum_{t=1}^{T} u(\mathbf{z}_t, \pi^{(\mathcal{E})}(\mathbf{z}_t), b_{f_t}(\mathbf{z}_t, \pi^{(\mathcal{E})}(\mathbf{z}_t))) - u(\mathbf{z}_t, \pi^{(\boldsymbol{\omega}^*)}(\mathbf{z}_t), b_{f_t}(\mathbf{z}_t, \pi^{(\boldsymbol{\omega}^*)}(\mathbf{z}_t)))\right] + 1$$

To conclude, it suffices to bound the discretization error, as

$$\mathbb{E}_{\mathbf{z}_1,\ldots,\mathbf{z}_T}\left[\sum_{t=1}^{T} u(\mathbf{z}_t, \pi^{(\boldsymbol{\omega}^*)}(\mathbf{z}_t), b_{f_t}(\mathbf{z}_t, \pi^{(\boldsymbol{\omega}^*)}(\mathbf{z}_t))) - u(\mathbf{z}_t, \pi_t(\mathbf{z}_t), b_{f_t}(\mathbf{z}_t, \pi_t(\mathbf{z}_t)))\right] \leq O\left(\sqrt{T\log|\Pi|}\right),$$

which follows from applying the standard regret guarantee of Hedge (Lemma B.1 in the Appendix). Applying Lemma 4.10,

$$\mathbb{E}_{\mathbf{z}_1,\ldots,\mathbf{z}_T}\left[\sum_{t=1}^{T} u(\mathbf{z}_t, \pi^{(\mathcal{E})}(\mathbf{z}_t), b_{f_t}(\mathbf{z}_t, \pi^{(\mathcal{E})}(\mathbf{z}_t))) - u(\mathbf{z}_t, \pi^{(\boldsymbol{\omega}^*)}(\mathbf{z}_t), b_{f_t}(\mathbf{z}_t, \pi^{(\boldsymbol{\omega}^*)}(\mathbf{z}_t)))\right]$$

$$= \sum_{i=1}^{K}(\mathbb{E}_{\mathbf{z}}[u(\mathbf{z}, \pi^{(\mathcal{E})}(\mathbf{z}), b_{\alpha^{(i)}}(\mathbf{z}, \pi^{(\mathcal{E})}(\mathbf{z}))) - u(\mathbf{z}, \pi^{(\boldsymbol{\omega}^*)}(\mathbf{z}), b_{\alpha^{(i)}}(\mathbf{z}, \pi^{(\boldsymbol{\omega}^*)}(\mathbf{z})))])\left(\sum_{t=1}^{T} \mathbb{E}_{\mathbf{z}_1,\ldots,\mathbf{z}_{t-1}}[\mathbb{1}\{f_t = \alpha^{(i)}\}]\right)$$

$$\leq \sum_{i=1}^{K} \mathbb{E}_{\mathbf{z}}[u(\mathbf{z}, \pi^{(\mathcal{E})}(\mathbf{z}), b_{\alpha^{(i)}}(\mathbf{z}, \pi^{(\mathcal{E})}(\mathbf{z})))] \cdot \left(\sum_{t=1}^{T} \mathbb{E}_{\mathbf{z}_1,\ldots,\mathbf{z}_{t-1}}[\mathbb{1}\{f_t = \alpha^{(i)}\} - T \cdot \boldsymbol{\omega}^*[i]\right)$$

$$+ \sum_{i=1}^{K} \mathbb{E}_{\mathbf{z}}[u(\mathbf{z}, \pi^{(\boldsymbol{\omega}^*)}(\mathbf{z}), b_{\alpha^{(i)}}(\mathbf{z}, \pi^{(\boldsymbol{\omega}^*)}(\mathbf{z})))] \cdot \left(T \cdot \boldsymbol{\omega}^*[i] - \sum_{t=1}^{T} \mathbb{E}_{\mathbf{z}_1,\ldots,\mathbf{z}_{t-1}}[\mathbb{1}\{f_t = \alpha^{(i)}\}\right)$$

where the inequality follows from adding and subtracting $\sum_{i=1}^{K} \mathbb{E}_{\mathbf{z}}[u(\mathbf{z}, \pi^{(\mathcal{E})}(\mathbf{z}), b_{\alpha^{(i)}}(\mathbf{z}, \pi^{(\mathcal{E})}(\mathbf{z})))] \cdot T \cdot \boldsymbol{\omega}^*[i]$ and $\sum_{i=1}^{K} \mathbb{E}_{\mathbf{z}}[u(\mathbf{z}, \pi^{(\boldsymbol{\omega}^*)}(\mathbf{z}), b_{\alpha^{(i)}}(\mathbf{z}, \pi^{(\boldsymbol{\omega}^*)}(\mathbf{z})))] \cdot T \cdot \boldsymbol{\omega}^*[i]$. Finally, we can upper-bound the discretization error by

$$2\sum_{i=1}^{K}\left|T \cdot \boldsymbol{\omega}^*[i] - \sum_{t=1}^{T} \mathbb{E}_{\mathbf{z}_1,\ldots,\mathbf{z}_{t-1}}[\mathbb{1}\{f_t = \alpha^{(i)}\}]\right| \leq 2K$$

by using the fact that the sender's utility is bounded in $[0, 1]$. Piecing everything together and observing that $|\Pi| \leq T^K$ gives us the desired regret guarantee. $\qquad\square$

# C Appendix for Section 5: Extension to bandit feedback

## C.1 Stochastic follower types and adversarial contexts

Recall from Section 4.1 that $\mathbb{1}_{(\sigma(\mathbf{x})=a_f)} \in \{0,1\}^K$ is the indicator vector whose $i$-th component is $\mathbb{1}\{\sigma^{(\mathbf{x})}(\alpha^{(i)}) = a_f\}$, i.e. the indicator that a follower of type $\alpha^{(i)}$ best-responds to mixed strategy $\mathbf{x}$ by playing action $a_f$. For any fixed policy $\pi$, we can write the sender's expected utility in round $t$ as

$$\mathbb{E}_{f_t}\left[u(\mathbf{z}_t, \pi(\mathbf{z}_t), b_{f_t}(\pi(\mathbf{z}_t)))\right] = \sum_{a_f \in \mathcal{A}_f} u(\mathbf{z}_t, \pi(\mathbf{z}_t), a_f) \cdot p(b_f(\pi(\mathbf{z}_t)) = a_f)$$

$$= \sum_{a_f \in \mathcal{A}_f} u(\mathbf{z}_t, \pi(\mathbf{z}_t), a_f) \cdot p(\mathbb{1}_{(\sigma(\pi(\mathbf{z}_t))=a_f)})$$

where $p(b_f(\pi(\mathbf{z})) = a_f) := \mathbb{E}_{f \sim \mathcal{F}}[\mathbb{1}\{b_f(\mathbf{z}) = a_f\}]$, the first line follows from the assumption of a non-adaptive adversary, the second line follows from the fact that $f_1, \dots, f_T$ are drawn i.i.d., and

$$p(b_f(\pi(\mathbf{z})) = a_f) = p(\mathbb{1}_{(\sigma(\pi(\mathbf{z}))=a_f)}) := \sum_{i=1}^{K} \mathbb{1}\{b_{\alpha^{(i)}}(\pi(\mathbf{z})) = a_f\} \cdot \mathbb{P}(f = \alpha^{(i)}),$$

where $\mathbb{P}(f = \alpha^{(i)})$ is the probability that follower $f$ is of type $\alpha^{(i)}$. Note that $p(\mathbb{1}_{(\sigma(\pi(\mathbf{z}_t))=a_f)})$ (and therefore $\mathbb{E}_{f_t}[u(\mathbf{z}_t, \pi(\mathbf{z}_t), b_{f_t}(\pi(\mathbf{z}_t)))]$) is linear in $\mathbb{1}_{(\sigma(\pi(\mathbf{z}_t))=a_f)}$.

Given this reformulation, a natural approach is to estimate $p(\mathbb{1}_{(\sigma(\pi(\mathbf{z}_t))=a_f)})$ as $\widehat{p}(\mathbb{1}_{(\sigma(\pi(\mathbf{z}_t))=a_f)})$ and act greedily with respect to our estimate, like we did in Section 4.1. To do so, we define the set $\mathcal{W} := \{\mathbb{1}_{(\sigma=a_f)} \mid \forall a_f \in \mathcal{A}_f, \ \sigma \in \Sigma\}$ and estimate $p(\mathbf{b})$ for every element $\mathbf{b}$ in the barycentric spanner $\mathcal{B} := \{\mathbf{b}^{(1)}, \dots, \mathbf{b}^{(K)}\}$ of $\mathcal{W}$.[12]

We estimate $p(\mathbf{b})$ as follows: For every $\mathbf{b} \in \mathcal{B}$, there must be a mixed strategy $\mathbf{x}^{(\mathbf{b})}$ and follower action $a^{(\mathbf{b})}$ such that $\mathbf{b} = \mathbb{1}_{(\sigma^{\mathbf{x}^{(b)}}=a^{(\mathbf{b})})}$. Therefore, if the leader plays mixed strategy $\mathbf{x}^{(\mathbf{b})}$ $N$ times, we set $\widehat{p}(\mathbf{b}) = \frac{1}{N} \sum_{t \in [N]} \mathbb{1}\{b_{f_t}(\mathbf{x}^{(\mathbf{b})}) = a^{(\mathbf{b})}\}$. Given estimates $\{\widehat{\mathbf{p}}(\mathbf{b})\}_{\mathbf{b} \in \mathcal{B}}$, we can estimate $p(\mathbb{1}_{(\sigma(\mathbf{x})=a_f)})$ for any $\mathbf{x} \in \mathcal{E}$ and $a_f \in \mathcal{A}_f$ as

$$\widehat{p}(\mathbb{1}_{(\sigma(\mathbf{x})=a_f)}) := \sum_{i=1}^{K} \lambda_i(\mathbb{1}_{(\sigma(\mathbf{x})=a_f)}) \cdot \widehat{p}(\mathbf{b}^{(i)}),$$

where $\lambda_i(\mathbb{1}_{(\sigma(\mathbf{x})=a_f)}) \in [-1,1]$ for $i \in [K]$ are the coefficients from the barycentric spanner.[13] Note that this is an unbiased estimator, due to the fact that $p(\mathbb{1}_{(\sigma(\mathbf{x})=a_f)})$ is a linear function.

Algorithm 3 plays each mixed strategy $\mathbf{x}^{(\mathbf{b})}$ for $\mathbf{b} \in \mathcal{B}$ $N > 0$ times in order to obtain an estimate of each $\mathbf{p}(\mathbf{b})$.[14] It then uses these estimates to construct estimates for all $\mathbb{1}_{(\sigma(\mathbf{x})=a_f)}$ (and therefore also $\mathbb{E}_f[u(\mathbf{z}, \mathbf{x}, b_f(\mathbf{x}))]$ for all $\mathbf{x} \in \mathcal{E}$ and $\mathbf{z} \in \mathcal{Z}$). Finally, in the remaining rounds Algorithm 3 acts greedily with respect to its estimate, much like in Algorithm 1.

**Theorem C.1.** *If $N = O\left(\frac{A_f^{2/3} T^{2/3} \log^{1/3}(T)}{K^{1/3}}\right)$, then the expected contextual Stackelberg regret of Algorithm 3 (Definition 4.5) satisfies*

$$\mathbb{E}[R(T)] = O\left(K^{2/3} A_f^{2/3} T^{2/3} \log^{1/3}(T)\right).$$

*Proof Sketch.* The key step in our analysis is to show that for any best-response function $\sigma \in \Sigma$ and follower action $a_f \in \mathcal{A}_f$, $\mathrm{Var}(\widehat{p}(\mathbb{1}_{(\sigma=a_f)})) \leq \frac{K}{N}$ (Lemma C.3). Using this fact, we can bound the cumulative total variation distance between $\widehat{p}(\mathbb{1}_{(\sigma(\mathbf{x}_t)=a_f)})$ and $p(\mathbb{1}_{(\sigma(\mathbf{x}_t)=a_f)})$ for any sequence of mixed strategies and follower actions in the "exploit" phase (Lemma C.5). The rest of the analysis follows similarly to the proof of Corollary 4.7. $\square$

---

[12]See Section 6.3 of Balcan et al. [5] for details on how to compute this barycentric spanner.

[13]For more details, see Proposition 2.2 in Awerbuch and Kleinberg [4].

[14]In other words, we ignore the context in the "explore" rounds.

## C.2 Stochastic contexts and adversarial follower types

We now turn our attention to learning under bandit feedback when the sequence of contexts is chosen stochastically and the choice of follower type is adversarial. While we still use barycentric spanners to estimate $\{\{\widehat{p}(\mathbb{1}_{(\sigma(\mathbf{x})=a_f)})\}_{a_f \in \mathcal{A}_f}\}_{\mathbf{x} \in \mathcal{E}}$, we can no longer do all of our exploration "up front" like in Appendix C.1 because the follower types are now adversarially chosen. Instead, we follow the technique of Balcan et al. [5] and split the time horizon into $Z$ consecutive, evenly-sized blocks. In block $B_\tau$, we pick a random time-step to estimate $p_\tau(\mathbb{1}_{(\sigma(\mathbf{x}^{(\mathbf{b})})=a^{(\mathbf{b})})})$, i.e. the probability that a follower in block $B_\tau$ best-responds to mixed strategy $\mathbf{x}^{(\mathbf{b})}$ by playing action $a^{(\mathbf{b})}$, for every element in our barycentric spanner $\mathcal{B}$. If whenever the leader plays $\mathbf{x}^{(\mathbf{b})}$ the follower best-responds with action $a^{(\mathbf{b})}$, we set $\widehat{p}_\tau(\mathbb{1}_{(\sigma(\mathbf{x}^{(\mathbf{b})})=a^{(\mathbf{b})})}) = 1$. Otherwise we set $\widehat{p}_\tau(\mathbb{1}_{(\sigma(\mathbf{x}^{(\mathbf{b})})=a^{(\mathbf{b})})}) = 0$. Since the time-step in which we play $\mathbf{x}^{(\mathbf{b})}$ is chosen uniformly from all time-steps in $B_\tau$, $\widehat{p}_\tau(\mathbb{1}_{(\sigma(\mathbf{x}^{(\mathbf{b})})=a^{(\mathbf{b})})})$ is an unbiased estimate of $p_\tau(\mathbb{1}_{(\sigma(\mathbf{x}^{(\mathbf{b})})=a^{(\mathbf{b})})})$. While $\widehat{p}_\tau(\mathbb{1}_{(\sigma(\mathbf{x}^{(\mathbf{b})})=a^{(\mathbf{b})})})$ no longer has low variance since it must be recomputed separately for every block $B_\tau$, it is still bounded. Therefore, we can use our estimates $\{\widehat{p}_\tau(\mathbb{1}_{(\sigma(\mathbf{x}^{(\mathbf{b})})=a^{(\mathbf{b})})})\}_{\mathbf{b} \in \mathcal{B}}$, along with the corresponding linear coefficients from the barycentric spanner, to get a bounded (and unbiased) estimate for every $p(\mathbb{1}_{(\sigma(\mathbf{x})=a_f)})$.

Once we have estimates for $\{\{p_\tau(\mathbb{1}_{(\sigma(\mathbf{x})=a_f)})\}_{a_f \in \mathcal{A}_f}\}_{\mathbf{x} \in \mathcal{E}}$, we proceed via a reduction to Algorithm 2. In particular, in every block $B_\tau$ we use our estimates $\{\{\widehat{p}_\tau(\mathbb{1}_{(\sigma(\mathbf{x})=a_f)})\}_{a_f \in \mathcal{A}_f}\}_{\mathbf{x} \in \mathcal{E}}$ to construct an (unbiased and bounded) estimate of the average utility for all policies in our finite policy class $\Pi$ during block $B_\tau$. At the end of each block, we feed this estimate into the Hedge update step, which updates the weights of all policies for the next block. Finally, when we are not exploring (i.e. estimating $p_\tau(\mathbb{1}_{(\sigma(\mathbf{x}^{(\mathbf{b})})=a^{(\mathbf{b})})})$ for some $\mathbf{b} \in \mathcal{B}$), we sample the leader's policy according to the distribution over policies given by Hedge from the previous block. This process is summarized in Algorithm 4.

**Theorem C.2.** *If $N = O(T^{2/3} A_f^{1/3} \log^{1/3} T)$, then Algorithm 4 obtains expected contextual Stackelberg regret (Definition 4.9)*

$$\mathbb{E}[R(T)] \leq O\left(KA_f^{1/3} T^{2/3} \log^{1/3}(T)\right).$$

*Proof Sketch.* The analysis proceeds similarly to the analysis of Theorem 6.1 in Balcan et al. [5]. We highlight the key differences here. The first key difference is that while Balcan et al. [5] play Hedge over a finite set of leader strategies, we play Hedge over a finite set of leader *policies*, each of which map to one of finitely-many leader strategies for a given context. Second, unlike in Balcan et al. [5] it is not sufficient to only estimate $\{\{p_\tau(\mathbb{1}_{(\sigma(\mathbf{x})=a_f)})\}_{a_f \in \mathcal{A}_f}\}_{\mathbf{x} \in \mathcal{E}}$ to obtain an unbiased estimate of the utility of each policy in $\Pi$ in each time block—we must also specify a context (or set of contexts) to use in our estimator. We show that it suffices to select a context uniformly at random from the contexts $\{\mathbf{z}_t\}_{t \in B_\tau}$ encountered in the block. $\qquad \square$

## C.3 Proofs for Appendix C.1: Stochastic follower types and adversarial contexts

**Lemma C.3.** *For any $\sigma_f \in \Sigma$ and $a_f \in \mathcal{A}_f$, $\mathrm{Var}(\widehat{p}(\mathbb{1}_{(\sigma=a_f)})) \leq \frac{K}{N}$.*

**Algorithm 3:** Learning with stochastic follower types: bandit feedback

---

Let $\mathcal{B} = \{\mathbf{b}^{(1)}, \ldots, \mathbf{b}^{(K)}\}$ be the Barycentric spanner of $\mathcal{W}$

**for** $i = 1, \ldots, K$ **do**

    **for** $\tau = 1, \ldots, N$ **do**

        Play mixed strategy $\mathbf{x}^{(\mathbf{b}^{(i)})}$, observe best-response $a_{f_{(i-1)\cdot N+\tau}}$

    **end**

    Compute $\widehat{p}(\mathbf{b}^{(i)}) = \frac{1}{N} \sum_{\tau=1}^{N} \mathbb{1}\{b_{f_\tau}(\mathbf{x}^{(\mathbf{b}^{(i)})}) = a^{(\mathbf{b}^{(i)})}\}$

**end**

Compute $\widehat{p}(\mathbb{1}_{(\sigma=a_f)}) = \sum_{i=1}^{K} \lambda_i(\mathbb{1}_{(\sigma=a_f)}) \cdot \widehat{p}(\mathbf{b}^{(i)})$ for all $\sigma \in \Sigma$, $a_f \in \mathcal{A}_f$

**for** $t = K \cdot N + 1, \ldots, T$ **do**

    Observe context $\mathbf{z}_t$, commit to mixed strategy

    $\mathbf{x}_t = \widehat{\pi}(\mathbf{z}_t) = \arg\max_{\mathbf{x} \in \mathcal{E}} \sum_{a_f \in \mathcal{A}_f} \widehat{p}(\mathbb{1}_{(\sigma(\mathbf{x})=a_f)}) \cdot u(\mathbf{z}_t, \mathbf{x}, a_f)$.

**end**

---

*Proof.*

$$
\begin{aligned}
\mathrm{Var}(\widehat{p}(\mathbb{1}_{(\sigma=a_f)})) &:= \mathbb{E}[(\widehat{p}(\mathbb{1}_{(\sigma=a_f)}))^2] - \mathbb{E}[\widehat{p}(\mathbb{1}_{(\sigma=a_f)})]^2 \\
&= \mathbb{E}[(\widehat{p}(\mathbb{1}_{(\sigma=a_f)}))^2] - p^2(\mathbb{1}_{(\sigma=a_f)}) \\
&= \mathbb{E}\left[\left(\sum_{j=1}^{K} \lambda_j(\mathbb{1}_{(\sigma=a_f)})\widehat{p}(\mathbf{b}^{(j)})\right)^2\right] - p^2(\mathbb{1}_{(\sigma=a_f)}) \\
&= \mathbb{E}\left[\sum_{j=1}^{K} \lambda_j^2(\mathbb{1}_{(\sigma=a_f)})\widehat{p}^2(\mathbf{b}^{(j)})\right. \\
&\quad + \left.\sum_{i=1}^{K}\sum_{j=1,j\neq i}^{K} \lambda_i(\mathbb{1}_{(\sigma=a_f)})\lambda_j(\mathbb{1}_{(\sigma=a_f)})\widehat{p}(\mathbf{b}^{(i)})\widehat{p}(\mathbf{b}^{(j)})\right] - p^2(\mathbb{1}_{(\sigma=a_f)}) \\
&= \sum_{j=1}^{K} \lambda_j^2(\mathbb{1}_{(\sigma=a_f)})\mathbb{E}[\widehat{p}^2(\mathbf{b}^{(j)})] \\
&\quad + \sum_{i=1}^{K}\sum_{j=1,j\neq i}^{K} \lambda_i(\mathbb{1}_{(\sigma=a_f)})\lambda_j(\mathbb{1}_{(\sigma=a_f)})\mathbb{E}[\widehat{p}(\mathbf{b}^{(i)})\widehat{p}(\mathbf{b}^{(j)})] - p^2(\mathbb{1}_{(\sigma=a_f)})
\end{aligned}
$$

Observe that since (1) the follower in each round is drawn independently from $\mathcal{F}$ and (2) the rounds used to compute $\widehat{p}(\mathbf{b}^{(i)})$ do not overlap with the rounds used to compute $\widehat{p}(\mathbf{b}^{(j)})$ for $j \neq i$, $\widehat{p}(\mathbf{b}^{(i)})$ and $\widehat{p}(\mathbf{b}^{(j)})$ are independent random variables for $j \neq i$. Therefore

$$
\begin{aligned}
\mathrm{Var}(\widehat{p}(\mathbb{1}_{(\sigma=a_f)})) &= \sum_{j=1}^{K} \lambda_j^2(\mathbb{1}_{(\sigma=a_f)})\mathbb{E}[\widehat{p}^2(\mathbf{b}^{(j)})] \\
&\quad + \sum_{i=1}^{K}\sum_{j=1,j\neq i}^{K} \lambda_i(\mathbb{1}_{(\sigma=a_f)})\lambda_j(\mathbb{1}_{(\sigma=a_f)})p(\mathbf{b}^{(i)})p(\mathbf{b}^{(j)}) - p^2(\mathbb{1}_{(\sigma=a_f)}).
\end{aligned}
$$

We now turn our focus to $\mathbb{E}[\widehat{p}^2(\mathbf{b}^{(j)})]$. Observe that

$$
\mathbb{E}[\widehat{p}^2(\mathbf{b}^{(j)})] = \mathbb{E}\left[\left(\frac{1}{N}\sum_{\tau=1}^{N}\mathbb{1}\{b_{f_\tau}(\mathbf{x}^{(\mathbf{b}^{(j)})} = a_f^{(\mathbf{b}^{(j)})})\}\right)^2\right]
$$

$$
= \frac{1}{N^2}\mathbb{E}\left[\sum_{\tau=1}^{N}\sum_{\tau'=1}^{N}\mathbb{1}\{b_{f_\tau}(\mathbf{x}^{(\mathbf{b}^{(j)})} = a_f^{(\mathbf{b}^{(j)})})\}\cdot\mathbb{1}\{b_{f_{\tau'}}(\mathbf{x}^{(\mathbf{b}^{(j)})} = a_f^{(\mathbf{b}^{(j)})})\}\right]
$$

$$
= \frac{1}{N^2}(N\cdot p(\mathbf{b}^{(j)}) + N(N-1)p^2(\mathbf{b}^{(j)}))
$$

Plugging this into our expression for $\mathrm{Var}(\widehat{p}(\mathbb{1}_{(\sigma=a_f)}))$, we see that

$$
\mathrm{Var}(\widehat{p}(\mathbb{1}_{(\sigma=a_f)})) = \frac{1}{N}\sum_{j=1}^{K}\lambda_j^2(\mathbb{1}_{(\sigma=a_f)})(p(\mathbf{b}^{(j)}) + (N-1)p^2(\mathbf{b}^{(j)}))
$$

$$
+ \sum_{i=1}^{K}\sum_{j=1,j\neq i}^{K}\lambda_i(\mathbb{1}_{(\sigma=a_f)})\lambda_j(\mathbb{1}_{(\sigma=a_f)})p(\mathbf{b}^{(i)})p(\mathbf{b}^{(j)}) - p^2(\mathbb{1}_{(\sigma=a_f)})
$$

$$
= \frac{1}{N}\sum_{j=1}^{K}\lambda_j^2(\mathbb{1}_{(\sigma=a_f)})(p(\mathbf{b}^{(j)}) - p^2(\mathbf{b}^{(j)}))
$$

$$
+ \sum_{i=1}^{K}\sum_{j=1}^{K}\lambda_i(\mathbb{1}_{(\sigma=a_f)})\lambda_j(\mathbb{1}_{(\sigma=a_f)})p(\mathbf{b}^{(i)})p(\mathbf{b}^{(j)}) - p^2(\mathbb{1}_{(\sigma=a_f)})
$$

$$
= \frac{1}{N}\sum_{j=1}^{K}\lambda_j(\mathbb{1}_{(\sigma=a_f)})p(\mathbf{b}^{(j)})\cdot\lambda_j(\mathbb{1}_{(\sigma=a_f)})(1 - p(\mathbf{b}^{(j)}))
$$

$$
+ \left(\sum_{j=1}^{K}\lambda_j(\mathbb{1}_{(\sigma=a_f)})p(\mathbf{b}^{(j)})\right)^2 - p^2(\mathbb{1}_{(\sigma=a_f)})
$$

$$
= \frac{1}{N}\sum_{j=1}^{K}\lambda_j(\mathbb{1}_{(\sigma=a_f)})p(\mathbf{b}^{(j)})\cdot\lambda_j(\mathbb{1}_{(\sigma=a_f)})(1 - p(\mathbf{b}^{(j)})) + p^2(\mathbb{1}_{(\sigma=a_f)}) - p^2(\mathbb{1}_{(\sigma=a_f)})
$$

$$
\leq \frac{K}{N}
$$

where the last line follows from the fact that $\lambda_j(\mathbb{1}_{(\sigma=a_f)}) \in [-1,1]$ and $p(\mathbf{b}^{(j)}) \in [0,1]$. $\qquad\square$

**Lemma C.4.** *For any* $\mathbf{z} \in \mathcal{Z}$,

$$
\sum_{a_f\in\mathcal{A}_f} p(\mathbb{1}_{(\sigma(\widehat{\pi}(\mathbf{z}))=a_f)})\cdot u(\mathbf{z},\widehat{\pi}(\mathbf{z}),a_f) \geq \sum_{a_f\in\mathcal{A}_f}\widehat{p}(\mathbb{1}_{(\sigma(\pi^{(\mathcal{E})}(\mathbf{z}))=a_f)})\cdot u(\mathbf{z},\pi^{(\mathcal{E})}(\mathbf{z}),a_f)
$$

$$
- \sum_{a_f\in\mathcal{A}_f}|\widehat{p}(\mathbb{1}_{(\sigma(\widehat{\pi}(\mathbf{z}))=a_f)}) - p(\mathbb{1}_{(\sigma(\widehat{\pi}(\mathbf{z}))=a_f)})|.
$$

*Proof.* By the definition of $\widehat{\pi}$,

$$\sum_{a_f \in \mathcal{A}_f} \widehat{p}(\mathbb{1}_{(\sigma(\pi^{(\mathcal{E})}(\mathbf{z}))=a_f)}) \cdot u(\mathbf{z}, \pi^{(\mathcal{E})}(\mathbf{z}), a_f) \leq \sum_{a_f \in \mathcal{A}_f} \widehat{p}(\mathbb{1}_{(\sigma(\widehat{\pi}(\mathbf{z}))=a_f)}) \cdot u(\mathbf{z}, \widehat{\pi}(\mathbf{z}), a_f)$$

$$= \sum_{a_f \in \mathcal{A}_f} p(\mathbb{1}_{(\sigma(\widehat{\pi}(\mathbf{z}))=a_f)}) \cdot u(\mathbf{z}, \widehat{\pi}(\mathbf{z}), a_f)$$

$$+ \sum_{a_f \in \mathcal{A}_f} (\widehat{p}(\mathbb{1}_{(\sigma(\widehat{\pi}(\mathbf{z}))=a_f)}) - p(\mathbb{1}_{(\sigma(\widehat{\pi}(\mathbf{z}))=a_f)})) \cdot u(\mathbf{z}, \widehat{\pi}(\mathbf{z}), a_f)$$

$$\leq \sum_{a_f \in \mathcal{A}_f} p(\mathbb{1}_{(\sigma(\widehat{\pi}(\mathbf{z}))=a_f)}) \cdot u(\mathbf{z}, \widehat{\pi}(\mathbf{z}), a_f)$$

$$+ \sum_{a_f \in \mathcal{A}_f} |\widehat{p}(\mathbb{1}_{(\sigma(\widehat{\pi}(\mathbf{z}))=a_f)}) - p(\mathbb{1}_{(\sigma(\widehat{\pi}(\mathbf{z}))=a_f)})|.$$

the desired result may be obtained by rearranging terms. $\qquad\square$

**Lemma C.5.** *For any sequence of mixed strategies* $\mathbf{x}_{NK+1}, \ldots, \mathbf{x}_T$,

$$\sum_{t=NK+1}^{T} \sum_{a_f \in \mathcal{A}_f} |\widehat{p}(\mathbb{1}_{(\sigma(\mathbf{x}_t)=a_f)}) - p(\mathbb{1}_{(\sigma(\mathbf{x}_t)=a_f)})| \leq 2A_f T \sqrt{\frac{K \log(T)}{N}}$$

*with probability at least* $1 - \frac{1}{T}$.

*Proof.* By Lemma C.3 and a Hoeffding bound, we have that

$$|\widehat{p}(\mathbb{1}_{(\sigma=a_f)}) - p(\mathbb{1}_{(\sigma=a_f)})| \leq \sqrt{\frac{2K \log(1/\delta)}{N}}$$

with probability at least $1 - \delta$, for any particular $(\sigma, a_f)$ pair. Taking a union bound over the randomness in estimating $p(\mathbf{b}^{(1)}), \ldots, p(\mathbf{b}^{(K)})$, we see that

$$|\widehat{p}(\mathbb{1}_{(\sigma=a_f)}) - p(\mathbb{1}_{(\sigma=a_f)})| \leq \sqrt{\frac{2K \log(K/\delta)}{N}}$$

with probability at least $1 - \delta$, simultaneously for all $(\sigma, a_f)$. The desired result follows by summing over $T$ and $A_f$, and setting $\delta = \frac{1}{T}$. $\qquad\square$

**Theorem C.1.** *If* $N = O\left(\frac{A_f^{2/3} T^{2/3} \log^{1/3}(T)}{K^{1/3}}\right)$, *then the expected contextual Stackelberg regret of Algorithm 3 (Definition 4.5) satisfies*

$$\mathbb{E}[R(T)] = O\left(K^{2/3} A_f^{2/3} T^{2/3} \log^{1/3}(T)\right).$$

*Proof.*

$$\mathbb{E}[R(T)] := \mathbb{E}_{f_1,\dots,f_T \sim \mathcal{F}}\left[\sum_{t=1}^{T} u(\mathbf{z}_t, \pi^*(\mathbf{z}_t), b_{f_t}(\pi^*(\mathbf{z}_t))) - u(\mathbf{z}_t, \pi_t(\mathbf{z}_t), b_{f_t}(\pi_t(\mathbf{z}_t)))\right]$$

$$\leq 1 + \mathbb{E}_{f_1,\dots,f_T \sim \mathcal{F}}\left[\sum_{t=1}^{T} u(\mathbf{z}_t, \pi^{(\mathcal{E})}(\mathbf{z}_t), b_{f_t}(\pi^{(\mathcal{E})}(\mathbf{z}_t))) - u(\mathbf{z}_t, \pi_t(\mathbf{z}_t), b_{f_t}(\pi_t(\mathbf{z}_t)))\right]$$

$$\leq 1 + KN + \mathbb{E}_{f_{KN+1},\dots,f_T \sim \mathcal{F}}\left[\sum_{t=KN+1}^{T} u(\mathbf{z}_t, \pi^{(\mathcal{E})}(\mathbf{z}_t), b_{f_t}(\pi^{(\mathcal{E})}(\mathbf{z}_t))) - u(\mathbf{z}_t, \widehat{\pi}(\mathbf{z}_t), b_{f_t}(\widehat{\pi}(\mathbf{z}_t)))\right]$$

$$= 1 + KN + \mathbb{E}_{f \sim \mathcal{F}}\left[\sum_{t=KN+1}^{T} u(\mathbf{z}_t, \pi^{(\mathcal{E})}(\mathbf{z}_t), b_f(\pi^{(\mathcal{E})}(\mathbf{z}_t))) - u(\mathbf{z}_t, \widehat{\pi}(\mathbf{z}_t), b_f(\widehat{\pi}(\mathbf{z}_t)))\right]$$

$$= 1 + KN + \mathbb{E}_{f \sim \mathcal{F}}\left[\sum_{t=KN+1}^{T} \sum_{a_f \in \mathcal{A}_f} u(\mathbf{z}_t, \pi^{(\mathcal{E})}(\mathbf{z}_t), a_f) \cdot \mathbb{1}\{b_f(\pi^{(\mathcal{E})}(\mathbf{z}_t)) = a_f\}\right.$$

$$\left. - u(\mathbf{z}_t, \widehat{\pi}(\mathbf{z}_t), a_f) \cdot \mathbb{1}\{b_f(\widehat{\pi}(\mathbf{z}_t)) = a_f\}\right]$$

$$= 1 + KN + \sum_{t=NK+1}^{T} \sum_{a_f \in \mathcal{A}_f} u(\mathbf{z}_t, \pi^{(\mathcal{E})}(\mathbf{z}_t), a_f) \cdot p(\mathbb{1}_{(\sigma(\pi^{(\mathcal{E})}(\mathbf{z}_t)) = a_f)}) - u(\mathbf{z}_t, \widehat{\pi}(\mathbf{z}_t), a_f) \cdot p(\mathbb{1}_{(\sigma(\widehat{\pi}(\mathbf{z}_t)) = a_f)})$$

By Lemma C.4,

$$\mathbb{E}[R(T)] \leq 1 + KN + \sum_{t=NK+1}^{T} \sum_{a_f \in \mathcal{A}_f} u(\mathbf{z}_t, \pi^{(\mathcal{E})}(\mathbf{z}_t), a_f)(p(\mathbb{1}_{(\sigma(\pi^{(\mathcal{E})}(\mathbf{z}_t)) = a_f)}) - \widehat{p}(\mathbb{1}_{(\sigma(\pi^{(\mathcal{E})}(\mathbf{z}_t)) = a_f)}))$$

$$+ \sum_{t=NK+1}^{T} \sum_{a_f \in \mathcal{A}_f} |\widehat{p}(\mathbb{1}_{(\sigma(\widehat{\pi}(\mathbf{z}_t)) = a_f)}) - p(\mathbb{1}_{(\sigma(\widehat{\pi}(\mathbf{z}_t)) = a_f)})|$$

$$\leq 1 + KN + \sum_{t=NK+1}^{T} \sum_{a_f \in \mathcal{A}_f} |\widehat{p}(\mathbb{1}_{(\sigma(\pi^{(\mathcal{E})}(\mathbf{z}_t)) = a_f)}) - p(\mathbb{1}_{(\sigma(\pi^{(\mathcal{E})}(\mathbf{z}_t)) = a_f)})|$$

$$+ \sum_{t=NK+1}^{T} \sum_{a_f \in \mathcal{A}_f} |\widehat{p}(\mathbb{1}_{(\sigma(\widehat{\pi}(\mathbf{z}_t)) = a_f)}) - p(\mathbb{1}_{(\sigma(\widehat{\pi}(\mathbf{z}_t)) = a_f)})|$$

$$\leq 3 + KN + 4A_f T \sqrt{\frac{K \log(T)}{N}}$$

where the last line follows from Lemma C.5. The desired result follows by the setting of $N$. $\qquad\square$

### C.4 Proofs for Appendix C.2: Stochastic contexts and adversarial follower types

**Definition C.6.** *Let* $u_\tau(\pi) := \sum_{a_f \in \mathcal{A}_f} u(\mathbf{z}_\tau, \pi(\mathbf{z}_\tau), a_f) \cdot p_\tau(\mathbb{1}_{(\sigma(\pi(\mathbf{z}_\tau)) = a_f)})$ *and* $\widehat{u}_\tau(\pi) :=$
$\sum_{a_f \in \mathcal{A}_f} u(\mathbf{z}_\tau, \pi(\mathbf{z}_\tau), a_f) \cdot \sum_{j=1}^{K} \lambda_j(\mathbb{1}_{(\sigma(\pi(\mathbf{z}_\tau)) = a_f)}) \cdot \widehat{p}_\tau(\mathbf{b}^{(j)})$, *where* $\mathbf{z}_\tau \sim \text{Unif}\{\mathbf{z}_t : t \in B_\tau\}$,
$\mathcal{B} = \{\mathbf{b}^{(1)}, \dots, \mathbf{b}^{(K)}\}$ *is the Barycentric spanner for* $\mathcal{W}$, *and* $\widehat{p}(\mathbf{b}) = 1$ *if* $b_{f_{t(\mathbf{b})}}(\mathbf{x}^{(\mathbf{b})}) = a_f^{(\mathbf{b})}$ *and*
$\widehat{p}(\mathbf{b}) = 0$ *otherwise.*

**Lemma C.7.** *For any fixed policy* $\pi$, $\mathbb{E}_{\{\mathbf{z}_\tau\}_{t \in B_\tau}} \mathbb{E}[\widehat{u}_\tau(\pi)] = \mathbb{E}_{\mathbf{z}_\tau \sim \mathcal{P}}[u_\tau(\pi)] =$
$\mathbb{E}_{\mathbf{z}_\tau \sim \mathcal{P}}[\sum_{a_f \in \mathcal{A}_f} u(\mathbf{z}_\tau, \pi(\mathbf{z}_\tau), a_f) \cdot p_\tau(\mathbb{1}_{(\sigma(\pi(\mathbf{z}_\tau)) = a_f)})]$, *where the second expectation is taken over
the randomness in selecting the explore time-steps and in drawing* $\mathbf{z}_\tau \sim \text{Unif}\{\mathbf{z}_t : t \in B_\tau\}$.
*Moreover,* $\widehat{u}_\tau(\pi) \in [-KA_f, KA_f]$.

**Algorithm 4:** Learning with stochastic contexts: bandit feedback

---

Consider $\Pi := \{\pi^{(\boldsymbol{\omega})}\}_{\boldsymbol{\omega} \in \Omega}$

Let $\mathbf{q}_1[\pi^{(\boldsymbol{\omega})}] := 1$, $\mathbf{p}_1[\pi^{(\boldsymbol{\omega})}] := \frac{1}{|\Pi|}$ for all $\pi^{(\boldsymbol{\omega})} \in \Pi$

Let $\mathcal{B} = \{\mathbf{b}^{(1)}, \ldots, \mathbf{b}^{(K)}\}$ be the Barycentric spanner of $\mathcal{W}$

**for** $\tau = 1, \ldots, Z$ **do**

    Choose random perturbation over $\mathcal{B}$ and explore time-steps in $B_\tau$ uniformly at random

    Choose a time-step in $B_\tau$ uniformly at random whose context will be used as $\mathbf{z}_\tau$

    **for** $t \in B_\tau$ **do**

        If $t$ is an explore time-step, play the corresponding mixed strategy $\mathbf{x}^{(\mathbf{b}_t)}$ in $\mathcal{B}$. If
$a_{f_t} = a^{(\mathbf{b}_t)}$, set $\widehat{p}_\tau(\mathbf{b}_t) = 1$. Otherwise, set $\widehat{p}_\tau(\mathbf{b}_t) = 0$.

        Otherwise ($t$ is an exploit round), sample $\pi_t \sim \mathbf{p}_t$, $a_{l,t} \sim \pi_t(\mathbf{z}_t)$.

    **end**

    For each policy $\pi^{(\boldsymbol{\omega})} \in \Pi$, compute

    $\widehat{\ell}_\tau[\pi^{(\boldsymbol{\omega})}] := - \sum_{a_f \in \mathcal{A}_f} \sum_{i=1}^K \lambda_i(\mathbb{1}_{(\sigma(\pi(\mathbf{z}_\tau)) = a_f)}) \cdot \widehat{p}(\mathbf{b}^{(i)}) \cdot u(\mathbf{z}_\tau, \pi^{(\boldsymbol{\omega})}(\mathbf{z}_\tau), a_f)$.

    Set $\mathbf{q}_{\tau+1}[\pi^{(\boldsymbol{\omega})}] = \exp\left(-\eta \sum_{s=1}^\tau \widehat{\ell}_s[\pi^{(\boldsymbol{\omega})}]\right)$ and

    $\mathbf{p}_{t+1}[\pi^{(\boldsymbol{\omega})}] = \mathbf{q}_{t+1}[\pi^{(\boldsymbol{\omega})}] / \sum_{\pi^{(\boldsymbol{\omega}')} \in \Pi} \mathbf{q}_{t+1}[\pi^{(\boldsymbol{\omega}')}]$.

**end**

---

*Proof.*

$$
\begin{aligned}
\mathbb{E}_{\{\mathbf{z}_t\}_{t \in B_\tau}} \mathbb{E}[\widehat{u}_\tau(\pi)] &= \mathbb{E}_{\{\mathbf{z}_t\}_{t \in B_\tau}} \mathbb{E}\left[\sum_{a_f \in \mathcal{A}_f} u(\mathbf{z}_\tau, \pi(\mathbf{z}_\tau), a_f) \cdot \sum_{j=1}^K \lambda_j(\mathbb{1}_{(\sigma(\pi(\mathbf{z}_\tau)) = a_f)}) \cdot \widehat{p}_\tau(\mathbf{b}^{(j)})\right] \\
&= \mathbb{E}_{\{\mathbf{z}_t\}_{t \in B_\tau}} \mathbb{E}_{\mathbf{z}_\tau \sim \mathrm{Unif}\{\mathbf{z}_t : t \in B_\tau\}}\left[\sum_{a_f \in \mathcal{A}_f} u(\mathbf{z}_\tau, \pi(\mathbf{z}_\tau), a_f) \cdot \sum_{j=1}^K \lambda_j(\mathbb{1}_{(\sigma(\pi(\mathbf{z}_\tau)) = a_f)}) \cdot \mathbb{E}[\widehat{p}_\tau(\mathbf{b}^{(j)})]\right] \\
&= \mathbb{E}_{\{\mathbf{z}_t\}_{t \in B_\tau}} \mathbb{E}_{\mathbf{z}_\tau \sim \mathrm{Unif}\{\mathbf{z}_t : t \in B_\tau\}}\left[\sum_{a_f \in \mathcal{A}_f} u(\mathbf{z}_\tau, \pi(\mathbf{z}_\tau), a_f) \cdot \sum_{j=1}^K \lambda_j(\mathbb{1}_{(\sigma(\pi(\mathbf{z}_\tau)) = a_f)}) \cdot p_\tau(\mathbf{b}^{(j)})\right] \\
&= \mathbb{E}_{\{\mathbf{z}_t\}_{t \in B_\tau}} \mathbb{E}_{\mathbf{z}_\tau \sim \mathrm{Unif}\{\mathbf{z}_t : t \in B_\tau\}}\left[\sum_{a_f \in \mathcal{A}_f} u(\mathbf{z}_\tau, \pi(\mathbf{z}_\tau), a_f) \cdot p_\tau(\mathbb{1}_{(\sigma(\pi(\mathbf{z}_\tau)) = a_f)})\right] \\
&= \mathbb{E}_{\mathbf{z}_\tau \sim \mathcal{P}}\left[\sum_{a_f \in \mathcal{A}_f} u(\mathbf{z}_\tau, \pi(\mathbf{z}_\tau), a_f) \cdot p_\tau(\mathbb{1}_{(\sigma(\pi(\mathbf{z}_\tau)) = a_f)})\right] = \mathbb{E}_{\mathbf{z}_\tau \sim \mathcal{P}}[u_\tau(\pi)]
\end{aligned}
$$

$\square$

The following lemma is analogous to Equation (1) in Balcan et al. [5].

**Lemma C.8.**

$$
\mathbb{E}_{\mathbf{z}_1, \ldots, \mathbf{z}_N \sim \mathcal{P}}\left[\sum_{\tau=1}^N u_\tau(\pi^{(\mathcal{E})}) - \mathbb{E} u_\tau(\pi_\tau)\right] \leq \sqrt{N\kappa \log |\Pi|}
$$

*where $R_{N,\kappa}$ is an upper-bound on the regret of (full-information) Hedge which takes as input a sequence of $N$ losses/utilities which are bounded in $[-\kappa, \kappa]$ and are parameterized by $\mathbf{z}_1, \ldots, \mathbf{z}_N$.*

*Proof.*

$$\mathbb{E}_{\mathbf{z}_1,\ldots,\mathbf{z}_N\sim\mathcal{P}}\left[\sum_{\tau=1}^{N}\sum_{\pi\in\Pi}\mathbf{p}_\tau[\pi]\cdot u_\tau(\pi)\right] = \mathbb{E}_{\mathbf{z}_1,\ldots,\mathbf{z}_N\sim\mathcal{P}}\left[\sum_{\tau=1}^{N}\sum_{\pi\in\Pi}\mathbf{p}_\tau[\pi]\cdot\mathbb{E}[\widehat{u}_\tau(\pi)]\right]$$

$$= \mathbb{E}_{\mathbf{z}_1,\ldots,\mathbf{z}_N\sim\mathcal{P}}\mathbb{E}\left[\sum_{\tau=1}^{N}\sum_{\pi\in\Pi}\mathbf{p}_\tau[\pi]\cdot\widehat{u}_\tau(\pi)\right]$$

$$\geq \mathbb{E}_{\mathbf{z}_1,\ldots,\mathbf{z}_N\sim\mathcal{P}}\mathbb{E}\left[\max_{\pi\in\Pi}\sum_{\tau=1}^{N}\widehat{u}_\tau(\pi) - R_{N,\kappa}\right]$$

$$\geq \mathbb{E}_{\mathbf{z}_1,\ldots,\mathbf{z}_N\sim\mathcal{P}}\left[\max_{\pi\in\Pi}\mathbb{E}\left[\sum_{\tau=1}^{N}\widehat{u}_\tau(\pi)\right] - R_{N,\kappa}\right]$$

$$= \mathbb{E}_{\mathbf{z}_1,\ldots,\mathbf{z}_N\sim\mathcal{P}}\left[\max_{\pi\in\Pi}\sum_{\tau=1}^{N}u_\tau(\pi) - R_{N,\kappa}\right],$$

where the first line uses Lemma C.7 and the fact that $\mathbf{z}_1,\ldots,\mathbf{z}_T\sim\mathcal{P}$ are i.i.d., and $R_{N,\kappa}$ is the regret of Hedge after $N$ time-steps when losses are bounded in $[-\kappa,\kappa]$. Rearranging terms and using the fact that the expected regret of Hedge after $N$ time-steps is at most $\sqrt{N\kappa\log|\Pi|}$ gets us the desired result. $\qquad\square$

**Theorem C.2.** *If $N = O(T^{2/3}A_f^{1/3}\log^{1/3}T)$, then Algorithm 4 obtains expected contextual Stackelberg regret (Definition 4.9)*

$$\mathbb{E}[R(T)] \leq O\left(KA_f^{1/3}T^{2/3}\log^{1/3}(T)\right).$$

*Proof.*

$$\mathbb{E}[R(T)] := \mathbb{E}\mathbb{E}_{\mathbf{z}_1,\ldots,\mathbf{z}_T\sim\mathcal{P}}\left[\sum_{t=1}^{T}u(\mathbf{z}_t,\pi^*(\mathbf{z}_t),b_{f_t}(\pi^*(\mathbf{z}_t))) - u(\mathbf{z}_t,\pi_t(\mathbf{z}_t),b_{f_t}(\pi_t(\mathbf{z}_t)))\right]$$

$$\leq 1 + \mathbb{E}\mathbb{E}_{\mathbf{z}_1,\ldots,\mathbf{z}_T\sim\mathcal{P}}\left[\sum_{t=1}^{T}u(\mathbf{z}_t,\pi^{(\mathcal{E})}(\mathbf{z}_t),b_{f_t}(\pi^{(\mathcal{E})}(\mathbf{z}_t))) - u(\mathbf{z}_t,\pi_t(\mathbf{z}_t),b_{f_t}(\pi_t(\mathbf{z}_t)))\right]$$

$$= 1 + \mathbb{E}\mathbb{E}_{\mathbf{z}_1,\ldots,\mathbf{z}_T\sim\mathcal{P}}\left[\sum_{\tau=1}^{N}\sum_{t\in B_\tau}u(\mathbf{z}_t,\pi^{(\mathcal{E})}(\mathbf{z}_t),b_{f_t}(\pi^{(\mathcal{E})}(\mathbf{z}_t))) - u(\mathbf{z}_t,\pi_t(\mathbf{z}_t),b_{f_t}(\pi_t(\mathbf{z}_t)))\right]$$

$$\leq 1 + KN + \mathbb{E}\mathbb{E}_{\mathbf{z}_1,\ldots,\mathbf{z}_T\sim\mathcal{P}}\left[\sum_{\tau=1}^{N}\sum_{t\in B_\tau}u(\mathbf{z}_t,\pi^{(\mathcal{E})}(\mathbf{z}_t),b_{f_t}(\pi^{(\mathcal{E})}(\mathbf{z}_t))) - u(\mathbf{z}_t,\pi_\tau(\mathbf{z}_t),b_{f_t}(\pi_\tau(\mathbf{z}_t)))\right]$$

$$= 1 + KN + \mathbb{E}\mathbb{E}_{\mathbf{z}_1,\ldots,\mathbf{z}_T \sim \mathcal{P}} \left[ \sum_{\tau=1}^{N} \sum_{t \in B_\tau} \sum_{a_f \in \mathcal{A}_f} u(\mathbf{z}_t, \pi^{(\mathcal{E})}(\mathbf{z}_t), a_f) \cdot \mathbb{1}\{b_{f_t}(\pi^{(\mathcal{E})}(\mathbf{z}_t)) = a_f\} \right.$$

$$\left. - u(\mathbf{z}_t, \pi_\tau(\mathbf{z}_t), a_f) \cdot \mathbb{1}\{b_{f_t}(\pi_\tau(\mathbf{z}_t)) = a_f\} \right]$$

$$= 1 + KN + \mathbb{E}\sum_{\tau=1}^{N} \mathbb{E}_{\mathbf{z}_1,\ldots,\mathbf{z}_{(\tau-1)\cdot|B_{\tau-1}|}} \mathbb{E}_{\mathbf{z}_t : t \in B_\tau | \mathbf{z}_1,\ldots,\mathbf{z}_{(\tau-1)\cdot|B_{\tau-1}|}} \left[ \sum_{t \in B_\tau} \sum_{a_f \in \mathcal{A}_f} \right.$$

$$\left. u(\mathbf{z}_t, \pi^{(\mathcal{E})}(\mathbf{z}_t), a_f) \cdot \mathbb{1}\{b_{f_t}(\pi^{(\mathcal{E})}(\mathbf{z}_t)) = a_f\} - u(\mathbf{z}_t, \pi_\tau(\mathbf{z}_t), a_f) \cdot \mathbb{1}\{b_{f_t}(\pi_\tau(\mathbf{z}_t)) = a_f\} \right]$$

$$= 1 + KN + \mathbb{E}\sum_{\tau=1}^{N} \mathbb{E}_{\mathbf{z}_1,\ldots,\mathbf{z}_{(\tau-1)\cdot|B_{\tau-1}|} \sim \mathcal{P}} \mathbb{E}_{\mathbf{z}_\tau \sim \mathcal{P}|\mathbf{z}_1,\ldots,\mathbf{z}_{(\tau-1)\cdot|B_{\tau-1}|}} \left[ \sum_{t \in B_\tau} \sum_{a_f \in \mathcal{A}_f} \right.$$

$$\left. u(\mathbf{z}_\tau, \pi^{(\mathcal{E})}(\mathbf{z}_\tau), a_f) \cdot \mathbb{1}\{b_{f_t}(\pi^{(\mathcal{E})}(\mathbf{z}_\tau)) = a_f\} - u(\mathbf{z}_\tau, \pi_\tau(\mathbf{z}_\tau), a_f) \cdot \mathbb{1}\{b_{f_t}(\pi_\tau(\mathbf{z}_\tau)) = a_f\} \right]$$

$$= 1 + KN + \mathbb{E}\sum_{\tau=1}^{N} \mathbb{E}_{\mathbf{z}_1,\ldots,\mathbf{z}_{(\tau-1)\cdot|B_{\tau-1}|} \sim \mathcal{P}} \mathbb{E}_{\mathbf{z}_\tau \sim \mathcal{P}|\mathbf{z}_1,\ldots,\mathbf{z}_{(\tau-1)\cdot|B_{\tau-1}|}} \left[ \sum_{a_f \in \mathcal{A}_f} u(\mathbf{z}_\tau, \pi^{(\mathcal{E})}(\mathbf{z}_\tau), a_f) \right.$$

$$\left. \cdot \left( \sum_{t \in B_\tau} \mathbb{1}\{b_{f_t}(\pi^{(\mathcal{E})}(\mathbf{z}_\tau)) = a_f\} \right) - u(\mathbf{z}_\tau, \pi_\tau(\mathbf{z}_\tau), a_f) \cdot \left( \sum_{t \in B_\tau} \mathbb{1}\{b_{f_t}(\pi_\tau(\mathbf{z}_\tau)) = a_f\} \right) \right]$$

where the second line follows from Lemma 4.4, the third from splitting the time horizon into blocks, the fourth from loss due to exploration, the fifth due to reformulating the reward as a function of different follower actions, the sixth due to linearity of expectation, and the seventh line follows from the fact that (1) $\pi_\tau$ is independent of $\mathbf{z}_t$ for all $t \in B_\tau$ and (2) $\mathbf{z}_1,\ldots,\mathbf{z}_T$ are independent.

$$\mathbb{E}[R(T)] \leq 1 + KN + B\mathbb{E}\sum_{\tau=1}^{N} \mathbb{E}_{\mathbf{z}_1,\ldots,\mathbf{z}_{(\tau-1)\cdot|B_{\tau-1}|} \sim \mathcal{P}} \mathbb{E}_{\mathbf{z}_\tau \sim \mathcal{P}|\mathbf{z}_1,\ldots,\mathbf{z}_{(\tau-1)\cdot|B_{\tau-1}|}} \left[ \sum_{a_f \in \mathcal{A}_f} u(\mathbf{z}_\tau, \pi^{(\mathcal{E})}(\mathbf{z}_\tau), a_f) \right.$$

$$\left. \cdot \left( \frac{1}{|B_\tau|} \sum_{t \in B_\tau} \mathbb{1}\{b_{f_t}(\pi^{(\mathcal{E})}(\mathbf{z}_\tau) = a_f\}) \right) - u(\mathbf{z}_\tau, \pi_\tau(\mathbf{z}_\tau), a_f) \cdot \left( \frac{1}{|B_\tau|} \sum_{t \in B_\tau} \mathbb{1}\{b_{f_t}(\pi_\tau(\mathbf{z}_\tau)) = a_f\} \right) \right]$$

$$= 1 + KN$$

$$+ B\mathbb{E}\sum_{\tau=1}^{N} \mathbb{E}_{\mathbf{z}_1,\ldots,\mathbf{z}_{(\tau-1)\cdot|B_{\tau-1}|} \sim \mathcal{P}} \mathbb{E}_{\mathbf{z}_\tau \sim \mathcal{P}|\mathbf{z}_1,\ldots,\mathbf{z}_{(\tau-1)\cdot|B_{\tau-1}|}} \left[ \sum_{a_f \in \mathcal{A}_f} u(\mathbf{z}_\tau, \pi^{(\mathcal{E})}(\mathbf{z}_\tau), a_f) \cdot p_\tau(\mathbb{1}_{(\sigma(\pi^{(\mathcal{E})})=a_f)}) \right.$$

$$\left. - u(\mathbf{z}_\tau, \pi_\tau(\mathbf{z}_\tau), a_f) \cdot p_\tau(\mathbb{1}_{(\sigma(\pi_\tau(\mathbf{z}_\tau))=a_f)}) \right]$$

$$\leq 1 + KN + B\mathbb{E}\sum_{\tau=1}^{N} \mathbb{E}_{\mathbf{z}_1,\ldots,\mathbf{z}_{(\tau-1)\cdot|B_{\tau-1}|} \sim \mathcal{P}} \mathbb{E}_{\mathbf{z}_\tau \sim \mathcal{P}|\mathbf{z}_1,\ldots,\mathbf{z}_{(\tau-1)\cdot|B_{\tau-1}|}} [u_\tau(\pi^{(\mathcal{E})}) - u_\tau(\pi_\tau)]$$

$$= 1 + KN + B \cdot \mathbb{E}_{\mathbf{z}_1,\ldots,\mathbf{z}_N \sim \mathcal{P}} \left[ \sum_{\tau=1}^{N} u_\tau(\pi^{(\mathcal{E})}) - \mathbb{E}u_\tau(\pi_\tau) \right]$$

$$\leq 1 + KN + B \cdot \sqrt{NKA_f \log|\Pi|}$$

$$\leq 1 + KN + TK \cdot \sqrt{\frac{A_f \log(T)}{N}}$$

where the first line comes from multiplying and dividing by $|B_\tau|$, the second line comes from the definition of $p_\tau$, the third from the definition of $u_\tau$, the fourth follows from linearity of expectation and the fact that $\mathbf{z}_1,\ldots,\mathbf{z}_T$ are i.i.d., the fifth follows from applying Lemma C.8, and the sixth line follows from the definition of $B$ and the fact that $|\Pi| \leq N^K$. Setting $N$ gets us the final result. $\square$

