# OpenReview forum: "Regret Minimization in Stackelberg Games with Side Information"
_NeurIPS.cc/2024/Conference — NeurIPS 2024 poster_

### Official Review · Reviewer_zDTp · 2024-07-11

**Soundness:** 3
**Presentation:** 3
**Contribution:** 2
**Rating:** 6
**Confidence:** 4

**Summary:**

The paper examines online learning within Stackelberg games, incorporating additional contextual information. Specifically, at each round $t$, both the follower and the leader observe a shared context $z_t$, which impacts their respective utilities. The leader, who is also the learner in this online learning framework, then selects a mixed strategy $x_t$​ from its set of possible actions. Following this, the follower observes the leader's actions and chooses a best-response action based on both the leader's action and the given context. The learner's objective is to choose a sequence of probability distributions that maximizes overall utility. The authors evaluate the performance of no-regret algorithms using the concept of *"policy regret,"* aiming to achieve sublinear regret relative to the optimal *"fixed contextual policy"* that assigns a probability distribution to each context.

The authors present several positive and negative results:

1. When the context and the type of players are adversarially selected, there is no $o(T)$-regret algorithm.

2. When the types of players arrive according to a fixed probability distribution but the contexts are adversarially selected, the authors provide an $O(T^{1/2})$-regret algorithm.

3. When the contexts arrive according to a fixed probability distribution but the types of followers arrive according to a fixed probability distribution, the authors provide an $O(T^{1/2})$-regret algorithm.

4. Finally, the authors extend their results to the bandit case by providing O(T2/3)O(T^{2/3})O(T2/3)-regret algorithms.

**Strengths:**

I find the problem of online learning in Stackelberg games, where both the follower and the leader observe contextual side information, to be well-motivated and of significant interest to the game theory and learning community. The authors have thoroughly examined several aspects of the problem. Despite the negative results in the case of adversarially selected contexts and types, they provide positive results in the stochastic case. Furthermore, the results appear solid and present considerable technical interest.

**Weaknesses:**

The regret bounds for the bandit case are not tight. However I believe the paper provides interesting first results for an interesting problem.

**Questions:**

What are the main challenges of extending your results in infinite action games with convex structure?

**Limitations:**

Yes

---

> ### Author Rebuttal · Authors · 2024-08-05
>
> [*“The regret bounds for the bandit case are not tight. However I believe the paper provides interesting first results for an interesting problem.”*]
>
> We would like to point out that it is actually unclear whether or not an algorithm exists for the bandit settings which achieves better than O(T^{2/3}) regret. While we hypothesize that getting O(T^{1/2}) regret should be possible, the algorithmic ideas used to get O(T^{1/2}) regret in the online Stackelberg setting without side information do not extend to our setting in a straightforward way.
>
> [*“What are the main challenges of extending your results in infinite action games with convex structure?”*]
>
> Since the leader’s strategy is a probability distribution over actions, one could view the leader as picking from infinitely-many actions. If you are asking about how one would extend our results to settings in which the leader’s strategy is not a probability simplex but is instead some general convex set, the problem becomes more tricky. To generalize to this setting, one would need to be able to reason about how different follower types would best-respond to different leader strategies. One could then hope to leverage the structure of this best-response in order to derive a computationally-tractable algorithm (i.e. in a way analogous to how we use Lemma 4.4).

---

> > ### Comment · Reviewer_zDTp · 2024-08-09
> > **Reviewer's Response**
> >
> > I have read the authors' response and I plan to keep my score.

---

### Official Review · Reviewer_8oy3 · 2024-07-12

**Soundness:** 3
**Presentation:** 3
**Contribution:** 2
**Rating:** 5
**Confidence:** 3

**Summary:**

The paper studies an online Stackelberg game, where the leader plays with a different follower type in a different context in each time step. The paper takes an online learning approach to solving this problem. It first that given that the context space is infinite, it is not possible to achieve sublinear regret via a reduction to an online linear threasholding problem. Therefore, the authors consider relaxed cases, where either the context or the follower type is chosen stochastically in each round. In both cases, the authors present algorithms that guanrantee sublinear regret, both for the full feedback and bandit feedback settings.

**Strengths:**

(+) The model is well motivated.

(+) The paper is clear and well presented. All analyses look sound and rigorous.

(+) The results presented are very complete, covering all cases of the model and both positive and corresponding negative results.

**Weaknesses:**

(-) The main impossibility result appears to reply on the fact that the context space is infinite and the model is non-linear w.r.t. the context.

(-) The analysis of the relaxed cases (stochastic follower type, or stochastic context) looks fairly standard and the results are expected. So overall the paper is more like an application of existing techniques to a problem motivated by a new context.

**Questions:**

Would the impossiblity result change if the context space is finite, or if the players' utility functions are linear w.r.t. the context.

**Limitations:**

The authors didn't seem to have addressed this explicitly, but the work is theoretical anyway, so this is minor.

---

> ### Author Rebuttal · Authors · 2024-08-05
>
> [*“The main impossibility result appears to reply on the fact that the context space is infinite and the model is non-linear w.r.t. the context.” and “Would the impossiblity result change if the context space is finite, or if the players' utility functions are linear w.r.t. the context.”*]
>
> We would like to emphasize that most work on learning in contextual settings focuses on the setting where the context space is infinite or large. With that being said, our impossibility result should carry over to the setting where the number of contexts is finite, but exponential-in-T. (This is because the impossibility result for no-regret learning in online linear thresholding carries over to the setting where the adversary can only select from an exponentially-large grid of uniformly-spaced points in [0, 1].)
>
> We would like to clarify that the leader’s utility function is not non-linear with respect to the context in our impossibility result. While we allow non-linear relationships in all of our positive results, the leader and follower utility functions actually have no explicit dependence on the context in our lower bound (see line 518). The intuition for the lower bound is that while the leader’s utility may not explicitly depend on the context, the adversary can “hide” information in the context about the receiver’s type. This ability to “hide” information makes it hard for the leader to compete with the best-in-hindsight policy.
>
> [*“The analysis of the relaxed cases (stochastic follower type, or stochastic context) looks fairly standard and the results are expected. So overall the paper is more like an application of existing techniques to a problem motivated by a new context.”*]
>
> We would like to clarify that our positive results do not follow from existing techniques. We will update the paper to better explain the technical innovation based on the reviewer’s feedback. There are several reasons why this is not the case. The setting of Section 4.1 (and its bandit analogue) have not been studied in the literature on non-contextual Stackelberg games. As such, there is no existing technique to apply in this setting. Our algorithm in Section 4.2 plays Hedge over a finite set of policies, while previous work on non-contextual Stackelberg games plays Hedge over a finite set of mixed strategies. While the two ideas may seem similar at this level of abstraction, showing that it suffices to consider a finite set of policies (each of which map to a finite set of context-dependent actions) is non-trivial and requires bounding a discretization error which does not appear in the setting without side information. We will update the final version to better explain the technical innovation.
>
> [*[On limitations]: “The authors didn't seem to have addressed this explicitly, but the work is theoretical anyway, so this is minor.”*]
>
> We discuss (what we view as) the most important limitations of our work (allowing intermediate forms of adversary, better regret rates in the bandit setting) in the Conclusion as directions for future work. However, we would be happy to add further discussion on other limitations in our next revision.

---

### Official Review · Reviewer_YfNJ · 2024-07-12

**Soundness:** 3
**Presentation:** 3
**Contribution:** 3
**Rating:** 6
**Confidence:** 3

**Summary:**

The paper presents a study of Stackelberg games with contextual side information, impacting their strategies in a game theoretic setting. The authors introduce a framework for analyzing online Stackelberg games, where a leader faces a sequence of followers, and both or either sequences—contexts and follower types—can be adversarially chosen. The paper contributes by showing the limitations of traditional non-contextual strategies and offering new algorithms that can handle stochastic elements in either the context or the follower sequences.

**Strengths:**

- The paper addresses a novel aspect of Stackelberg games by incorporating side information, which is realistically present in many practical applications but often ignored in theoretical models.
- The paper is technically sound with rigorous proofs and a clear exposition of both theoretical and practical implications of the findings.
- The paper is well-written and organized. Concepts are introduced systematically, and the flow from problem statement to results is logical and easy to follow.

**Weaknesses:**

- The paper could improve by providing numerical experiments or case studies that demonstrate the efficacy of the proposed algorithms.
- The practical implications are clear for certain fields, but the paper could further elaborate on how these findings might influence other areas of research or industry applications.

**Questions:**

- The paper would definitely benefit from the addition of numerical experiments or case studies.
- Could the authors provide more clarity on how the adversarial model for context selection was validated? Are there empirical data or specific scenarios where this model reflects real-world conditions?
- How would the author compare the results with existing methods for handling contextual information in game theory, such as contextual bandits or online learning with expert advice? Can the authors comment on how their approach compares to existing methods in terms of computational efficiency and practical deployability in real systems?
- How does the proposed algorithm perform if the model of side information is mis-specified? For instance, if the actual distribution of contexts or follower types deviates significantly from the stochastic model assumed, what is the impact on the regret bounds?
- Several theoretical assumptions are crucial and well presented in the paper. How sensitive are the main results to these assumptions? If some of these assumptions might not hold, how would this affect the applicability of the results?
- The paper could benefit from a deeper discussion on the limitations regarding the scalability of the algorithms when the number of contexts or follower types is large. What are the computational implications?

**Limitations:**

The authors have not explicitly addressed limitations or potential negative societal impacts of their work. A clearer identification of potential limitations, such as dependency on the accurate modeling of side information and follower behavior, would strengthen the paper.

---

> ### Author Rebuttal · Authors · 2024-08-05
>
> [*“The paper would definitely benefit from the addition of numerical experiments or case studies.”*]
>
> We hope our numerical simulations address your concerns regarding the lack of experimental results.
>
> [*“Could the authors provide more clarity on how the adversarial model for context selection was validated? Are there empirical data or specific scenarios where this model reflects real-world conditions?”*]
>
> The common motivation for studying adversarial models in online learning is not that we believe that the data-generating process is adversarial, but that we believe it is not i.i.d. If an algorithm can handle adversarially-generated data, then it can also handle data which is not i.i.d. in some way, while being agnostic as to how the data is generated. In Stackelberg games with side information, there are many examples of settings in which the contexts would not be i.i.d. In the wildlife protection and patrol boat examples, weather conditions for each day are not sampled i.i.d. from some distribution; instead, there is some dependence from day-to-day.
>
> [*“How would the author compare the results with existing methods for handling contextual information in game theory, such as...”*]
>
> We would like to emphasize that existing methods for handling contextual information (e.g. contextual bandit algorithms, online learning with expert advice) are usually not studied in strategic settings where there is more than a single agent. In general, such single-agent methods would perform poorly in game-theoretic settings, since they do not take the other strategic agents into consideration when making decisions. While our problem may be viewed as a special case of online contextual adversarial learning, applying algorithms for this problem out-of-the-box to our setting generally results in worse regret rates and would require more stringent assumptions on the data-generating process than what we require. (See line 485 for more details.)
>
> Others have also studied various forms of side information in different game settings. We include a discussion on these works in Appendix A.
>
> [*“Can the authors comment on how their approach compares to existing methods in terms of computational efficiency...”*]
>
> The computational complexity of our algorithms essentially match those for no-regret learning in Stackelberg game settings without contexts (up to small differences in polynomial factors). While both contextual and non-contextual algorithms for no-regret learning in Stackelberg games have exponential per-round complexity, this is unavoidable due to an existing hardness result for the non-contextual setting (see Li et al. [23]). With that being said, we believe that studying special cases of both the contextual and non-contextual settings where efficient learning is possible is an interesting and important direction for future research.
>
> [*“How does the proposed algorithm perform if the model of side information is mis-specified? For instance...”*]
>
> Thanks for the interesting question. While our impossibility result of Section 3 rules out the ability to learn when the distributions are arbitrarily misspecified, no-regret learning may still be possible under other, intermediary, forms of adversary (such as the one you propose). We highlight this as an interesting direction for future work in Section 6 (Conclusion).
>
> [*“Several theoretical assumptions are crucial and well presented in the paper. How sensitive are the main results to these assumptions?”*]
>
> The assumptions made through Section 4 (e.g. known utility functions, follower being one of K types, finite leader/follower actions, full feedback) are standard in the literature on no-regret learning in (Stackelberg) games and are reasonable under many settings.
>
> If you are referring to the assumptions which are unique to Section 5, the results are mixed. We believe that our results in Section 5 could be extended to handle adaptive adversaries by using a more clever exploration strategy. However, our algorithmic ideas in Section 5 do not readily extend to the setting in which the follower’s utility also depends on the context. We view relaxing this assumption as an interesting direction for future research.
>
> [*“The paper could benefit from a deeper discussion on the limitations regarding the scalability of the algorithms when the number of contexts or follower types is large. What are the computational implications?”*]
>
> We are happy to add a deeper discussion on computational runtime in our next revision. In short, our runtimes have no dependence on the number of contexts, and we inherit the exponential-in-K runtime from the non-contextual Stackelberg game setting (where K is the number of follower types). The implication of this is that while our results scale well to settings with a large (or infinitely-many) number of contexts, the number of different follower types should be small/constant.
>
> [*“The authors have not explicitly addressed limitations or potential negative societal impacts of their work. A clearer identification of..”*]
>
> We discuss (what we view as) the most important limitations of our work (allowing intermediate forms of adversary, better regret rates in the bandit setting) in the Conclusion as directions for future work. However, we would be happy to add additional discussions on the topics you propose.
>
> We chose not to discuss the societal implications of our work because our results are largely theoretical. With that being said, we anticipate that any societal implications of our work will be positive, since algorithms for learning in Stackelberg games are usually deployed in socially-beneficial domains. For example, in airport security, patrol strategies for drug-sniffing dogs could be modified to take factors such as the time of year, airport congestion, etc. into consideration. In wildlife protection domains, park rangers’ patrol schedules can be informed by things like observed tire tracks, or the current weather conditions.

---

> ### Comment · Reviewer_YfNJ · 2024-08-11
> **Thank you for your response**
>
> I really appreciate your detailed response to my questions and additional experiments. After reading the authors' responses and other reviewers' comments, I will keep my score.

---

### Official Review · Reviewer_C9HM · 2024-07-15

**Soundness:** 4
**Presentation:** 3
**Contribution:** 3
**Rating:** 6
**Confidence:** 4

**Summary:**

This paper studied the regret minimization in Stackelberg games with side information which consider the additional information available to each player. The paper found that achieving no-regret learning is impossible in fully adversarial settings. However, it demonstrated that no-regret learning is achievable in scenarios where either the sequence of contexts or followers is chosen stochastically.

**Strengths:**

The idea of using side-information to learn to play Stackelberg games is interesting.
Compared to previous work on learning in Stackelberg games, this paper takes into consideration the additional information available to both the leader and followers at each round, which is more complicated and realistic.
The paper provide an impossibility result and also identifies a setting where no-regret learning is possible. An algorithm is also provided to achieve no-regret.

**Weaknesses:**

This paper lacks experimental results to verify the theoretical analysis.

**Questions:**

I do not have any questions for the authors.

**Limitations:**

The limitations have been addressed properly.

---

> ### Author Rebuttal · Authors · 2024-08-05
>
> [*”This paper lacks experimental results to verify the theoretical analysis.”*]
>
> We hope our numerical simulations address your concerns regarding the lack of experimental results.

---

> > ### Comment · Reviewer_C9HM · 2024-08-13
> >
> > Thank you for your response and the numerical simulation. I will keep my score.

---

### Author Rebuttal · Authors · 2024-08-05

Thanks for taking the time and effort to review our submission. To summarize, we initiate the study of Stackelberg games with side information, which despite the presence of side information in many Stackelberg game settings, has not received attention from the community so far. We provide algorithms for regret minimization in Stackelberg games with side information, whose analysis requires new ideas and techniques when compared to algorithms for learning in Stackelberg games without side information.

 At the request of reviewers C9HM and YfNJ, we have included numerical experiments (summarized in the attached pdf) which show that our algorithms compare favorably to algorithms for learning in Stackelberg games which do not take side information into consideration. If you would like to see our code, we are happy to provide an anonymized link to the AC (as per the NeurIPS rebuttal instructions). Please find our responses to your other questions below.

---

### Decision · Program_Chairs · 2024-09-25

**Decision:**

Accept (poster)

**Comment:**

This paper studied the regret minimization in Stackelberg games with side information which consider the additional information available to each player. The paper found that achieving no-regret learning is impossible in fully adversarial settings. However, it demonstrated that no-regret learning is achievable in scenarios where either the sequence of contexts or followers is chosen stochastically.

The idea of using side-information to learn to play Stackelberg games is interesting and well-motivated.
The paper is technically sound with rigorous proofs and a clear exposition of both theoretical and practical implications of the findings.
On the other side many doubts have been raised both on the practicality and applicability in real system of current results, and on the apparent difficulty of generalizing results to more general settings.
The authors should also revise the paper by including more detailed comments on the computational complexity, on the similarities and differences of the proposing method with contextual bandit learning, and possibly include numerical results, as highlighted in the reviews.